# Poisson Flow Generative Models

**Yilun Xu**[*], **Ziming Liu**[*], **Max Tegmark, Tommi Jaakkola**
Massachusetts Institute of Technology
{ylxu, zmliu, tegmark}@mit.edu; tommi@csail.mit.edu

## Abstract

We propose a new "Poisson flow" generative model (PFGM) that maps a uniform distribution on a high-dimensional hemisphere into any data distribution. We interpret the data points as electrical charges on the $z = 0$ hyperplane in a space augmented with an additional dimension $z$, generating a high-dimensional electric field (the gradient of the solution to Poisson equation). We prove that if these charges flow upward along electric field lines, their initial distribution in the $z = 0$ plane transforms into a distribution on the hemisphere of radius $r$ that becomes *uniform* in the $r \to \infty$ limit. To learn the bijective transformation, we estimate the normalized field in the augmented space. For sampling, we devise a backward ODE that is anchored by the physically meaningful additional dimension: the samples hit the (unaugmented) data manifold when the $z$ reaches zero. Experimentally, PFGM achieves current state-of-the-art performance among the normalizing flow models on CIFAR-10, with an Inception score of $9.68$ and a FID score of $2.35$. It also performs on par with the state-of-the-art SDE approaches while offering $10\times$ to $20\times$ acceleration on image generation tasks. Additionally, PFGM appears more tolerant of estimation errors on a weaker network architecture and robust to the step size in the Euler method. The code is available at https://github.com/Newbeeer/poisson_flow.

## 1 Introduction

Deep generative models are a prominent approach for data generation, and have been used to produce high quality samples in image [1], text [2] and audio [35], as well as improve semi-supervised learning [20], domain generalization [25] and imitation learning [15]. However, current deep generative models also have limitations, such as unstable training objectives (GANs [1, 12, 17]) and low sample quality (VAEs [21], normalizing flows [6]). New techniques [12, 24] are introduced to stablize the training of CNN-based or ViT-based GAN models. Although recent advances on diffusion [16] and scored-based models [33] achieve comparable sample quality to GAN's without adversarial training, these models have a slow stochastic sampling process. [33] proposes backward ODE samplers (normalizing flow) that speed up the sampling process but these methods have not yet performed on par with the SDE counterparts.

We present a new "Poisson flow" generative model (**PFGM**), exploiting a remarkable physics fact that generalizes to $N$ dimensions. As illustrated in Fig. 1(a), motion in a viscous fluid transforms any planar charge distribution into a uniform angular distribution. Specifically, we interpret $N$-dimensional data points $\mathbf{x}$ (images, say) as positive electric charges in the $z = 0$ plane of an $N + 1$-dimensional space (see Fig. 1(a)) filled with a viscous liquid (say honey). A positive charge with $z > 0$ will be repelled by the other charges and move in the direction of their repulsive force, eventually crossing an imaginary hemisphere of radius $r$. We show that, remarkably, if the the original charge distribution is let loose just above $z = 0$, this law of motion will cause a *uniform* distribution for their hemisphere crossings in the $r \to \infty$ limit.

---

[*]Equal Contribution.

36th Conference on Neural Information Processing Systems (NeurIPS 2022).

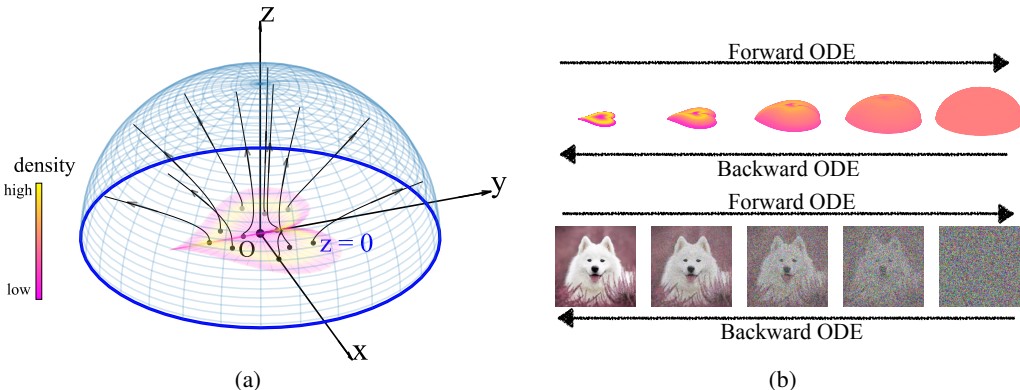

Figure 1: **(a)** 3D Poisson field trajectories for a heart-shaped distribution **(b)** The evolvements of a distribution (**top**) or an (augmented) sample (**bottom**) by the forward/backward ODEs pertained to the Poisson field.

Our Poisson flow generative process reverses the forward process: we generate a uniform distribution of negative charges on the hemisphere, then track their motion back to the $z = 0$ plane, where they will be distributed as the data distribution. A Poisson flow can be viewed as a type of continuous normalizing flows [4, 10, 33] in the sense that it continuously maps between an arbitrary distribution and an easily sampled one: in the previous works an $N$-dimensional Gaussian and in PFGM a uniform distribution on an $N$-dimensional hemisphere. In practice, we implement the Poisson flow by solving a pair of forward/backward ordinary differential equations (ODEs) induced by the electric field (Fig. 1(b)) given by the $N$-dimensional version of Coulomb's law (the gradient of the solution to the Poisson's equation with the data as sources). We will interchangeably refer to this gradient as the *Poisson field*, since electric fields normally refer to the special case $N = 3$.

The proposed generative model PFGM has a stable training objective and empirically outperforms previously state-of-the-art continuous flow methods [30, 33]. As a different iterative method, PFGM offers two advantages compared to score-based methods [32, 33]. First, the ODE process of PFGM achieves faster sampling speeds than the SDE samplers in [33]. while retaining comparable performance. Second, our backward ODE exhibits better generation performance than the reverse-time ODEs of VE/VP/sub-VP SDEs [33], as well as greater stability on a weaker architecture NSCNv2 [32]. The rationale for robustness is that the time variables in these ODE baselines are strongly correlated with the sample norms during training time, resulting in a less error-tolerant inference. In contrast, the tie between the anchored variable and the sample norm in PFGM is much weaker.

Experimentally, we show that PFGM achieves current state-of-the-art performance on CIFAR-10 dataset in the normalizing flow family, with FID/Inception scores of $2.48/9.65$ (w/ DDPM++ [33]) and $2.35/9.68$ (w/ DDPM++ deep [33]). It performs competitively with current state-of-the-art SDE samplers [33] and provides 10× to 20× speed up across datasets. Notably, the backward ODE in PFGM is the *only* ODE-based sampler that can produce decent samples on its own on NCSNv2 [32], while other ODE baselines fail without corrections. In addition, PFGM demonstrates the robustness to the step size in the Euler method, with a varying number of function evaluations (NFE) ranging from 10 to 100. We further showcase the utility of the invertible forward/backward ODEs of the Poisson field on likelihood evaluation and image manipulations, and its scalability to higher resolution images on LSUN bedroom 256 × 256 dataset.

## 2 Background and Related works

**Poisson equation** Let $\mathbf{x} \in \mathbb{R}^N$ and $\rho(\mathbf{x}) : \mathbb{R}^N \to \mathbb{R}$ be a *source* function. We assume that the source function has a compact support, $\rho \in \mathcal{C}^0$ and $N \geq 3$. The Poisson equation is

$$\nabla^2 \varphi(\mathbf{x}) = -\rho(\mathbf{x}), \tag{1}$$

where $\varphi(\mathbf{x}) : \mathbb{R}^N \to \mathbb{R}$ is called the *potential function*, and $\nabla^2 \equiv \sum_{i=1}^{N} \frac{\partial^2}{\partial x_i^2}$ is the Laplacian operator. It is usually helpful to define the gradient field $\mathbf{E}(\mathbf{x}) = -\nabla \varphi(\mathbf{x})$ and rewrite the Poisson equation as

$\nabla \cdot \mathbf{E} = \rho$, known in physics as Gauss's law [11]. The Poisson equation is widely used in physics, giving rise to Newton's gravitational theory [9] and the electrostatic theory [11], when $\rho(\mathbf{x})$ is interpreted as mass density or electric charge density, respectively. $\mathbf{E}$ is the $N$-dimensional analog of the electric field. The Poisson equation Eq. (1) (with zero boundary condition at infinity) admits a unique simple integral solution [2]:

$$\varphi(\mathbf{x}) = \int G(\mathbf{x}, \mathbf{y})\rho(\mathbf{y})d\mathbf{y}, \quad G(\mathbf{x}, \mathbf{y}) = \frac{1}{(N-2)S_{N-1}(1)} \frac{1}{\|\mathbf{x} - \mathbf{y}\|^{N-2}}, \qquad (2)$$

where $S_{N-1}(1)$ is a geometric constant representing the surface area of the unit $(N-1)$-sphere [3], and $G(\mathbf{x}, \mathbf{y})$ is the extension of Green's function in $N$-dimensional space (details in Appendix A.3). The negative gradient field of $\varphi(\mathbf{x})$, referred as *Poisson field* of the source $\rho$, is

$$\mathbf{E}(\mathbf{x}) = -\nabla\varphi(\mathbf{x}) = -\int \nabla_{\mathbf{x}} G(\mathbf{x}, \mathbf{y})\rho(\mathbf{y})d\mathbf{y}, \quad \nabla_{\mathbf{x}} G(\mathbf{x}, \mathbf{y}) = -\frac{1}{S_{N-1}(1)} \frac{\mathbf{x} - \mathbf{y}}{\|\mathbf{x} - \mathbf{y}\|^{N}}. \qquad (3)$$

Qualitatively, the Poisson field $\mathbf{E}(\mathbf{x})$ points away from sources, or equivalently $-\mathbf{E}(\mathbf{x})$ points towards sources, as illustrated in Fig. 1. It is straightforward to check that when $\rho(\mathbf{x}) \to \delta(\mathbf{x} - \mathbf{y})$, we get $\varphi(\mathbf{x}) \to G(\mathbf{x}, \mathbf{y})$ and $\mathbf{E}(\mathbf{x}) \to -\nabla_{\mathbf{x}} G(\mathbf{x}, \mathbf{y})$. This implies that $G(\mathbf{x}, \mathbf{y})$ and $-\nabla_{\mathbf{x}} G(\mathbf{x}, \mathbf{y})$ can be interpreted as the potential function and the gradient field generated by a unit point source, *e.g.*, a point charge, located at $\mathbf{y}$. When $\rho(\mathbf{x})$ takes general forms but has bounded support, simple asymptotics exist for $\|\mathbf{x}\| \gg \|\mathbf{y}\|$. To the lowest order, $\mathbf{E}(\mathbf{x}) = \nabla_{\mathbf{x}} G(\mathbf{x}, \mathbf{y})|_{\mathbf{y}=\mathbf{0}} \sim \mathbf{x}/\|\mathbf{x}\|^{N}$ behaves as if it were generated by a unit point source at $\mathbf{y} = 0$. In physics, the power law decay is considered to be long-range (compared to exponential decay) [11].

**Particle dynamics in a Poisson field** The Poisson field immediately defines a flow model, where the probability distribution evolves according to the gradient flow $\partial p_t(\mathbf{x})/\partial t = -\nabla \cdot (p_t(\mathbf{x})\mathbf{E}(\mathbf{x}))$. The gradient flow is a special case of the Fokker-Planck equation [28], where the diffusion coefficient is zero. Intuitively we can think of $p_t(\mathbf{x})$ as represented by a population of particles. The corresponding (non-diffusion) case of the Itô process is the forward ODE $\frac{d\mathbf{x}}{dt} = \mathbf{E}(\mathbf{x})$. We can interpret the trajectories of the ODE as particles moving according to the Poisson field $\mathbf{E}(x)$, with initial states drawn from $p_0$. The physical picture of the forward ODE is a charged particle under the influence of electric fields in the overdamped limit (details in Appendix F).

The dynamics is also *rescalable* in the sense that the particle trajectory remains the same for $\frac{d\mathbf{x}}{dt} = \pm f(\mathbf{x})\mathbf{E}(\mathbf{x})$ for $f(\mathbf{x}) > 0, f(\mathbf{x}) \in \mathcal{C}^1$, because the time rescaling $dt \to f(\mathbf{x}(t))dt$ recovers $\frac{d\mathbf{x}}{dt} = \pm\mathbf{E}(\mathbf{x})$. Note that the dynamics is stiff due to the power law factor in the denominator in Eq. (3), posing computational challenges. Luckily the rescalablility allows us to rescale $\mathbf{E}(\mathbf{x})$ properly to get new ODEs (formally defined later in Section 3.3) that are more amenable for sampling.

**Generative Modeling via ODE** Generative modeling can be done by transforming a base distribution to a data distribution via mappings defined by ODEs. The ODE-based samplers allow for adaptive sampling, exact likelihood evaluation and modeling of continuous-time dynamics [4, 33]. Previous works broadly fall into two lines. [4, 3] introduce a continuous-time normalizing flow model that can be trained with maximum likelihood by the instantaneous change-of-variables formula [4]. For sampling, they directly integrate the learned invertiable mapping over time. Another work [33] unifies the scored-based model [31, 32] and diffusion model [16] into a general diffusion process, and uses the reverse-time ODE of the diffusion process for sampling. They show that the reverse-time ODE produces high quality samples with improved architecture.

## 3 Poisson Flow Generative Models

In this section, we start with the properties of the Poisson flow in the augmented space and show how to draw samples from the data distribution by following the backward ODE of the Poisson flow (Section 3.1). We then discuss how to actually learn a normalized Poisson field from data samples through simulations of the forward ODE (Section 3.2) and present an equivalent backward ODE that allows for exponentially decay on $z$ (Section 3.3).

---

[2]Eq. (2) is valid for $N \geq 3$. When $N = 2$, the Green's function is $G(\mathbf{x}, \mathbf{y}) = -\log(\|\mathbf{x} - \mathbf{y}\|)/2\pi$. We assume $N \geq 3$ since $N$ is typically large in the relevant applications.

[3]The $N$-sphere with radius $r$ is defined as $\{\mathbf{x} \in \mathbb{R}^{N+1}, \|\mathbf{x}\| = r\}$

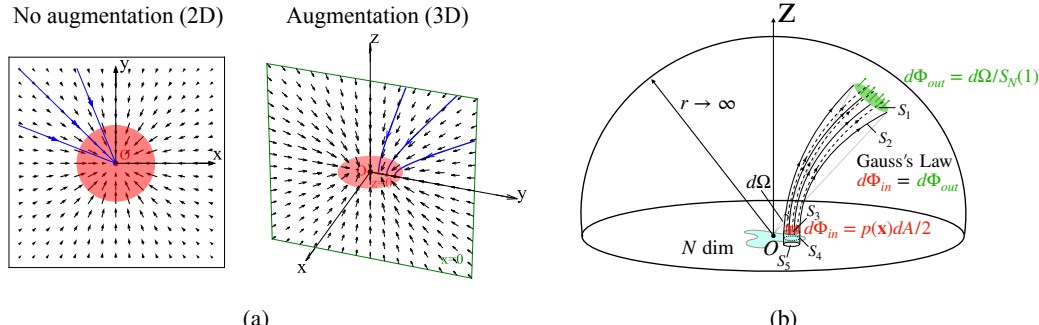

No augmentation (2D)  Augmentation (3D)

(a)             (b)

Figure 2: **(a)** Poisson field (black arrows) and particle trajectories (blue lines) of a 2D uniform disk (red). **Left** (no augmentation, 2D): all particles collapse to the disk center. **Right** (augmentation, 3D): particles hit different points on the disk. **(b)** Proof idea of Theorem 1. By Gauss's Law, the outflow flux $d\Phi_{out}$ equals the inflow flux $d\Phi_{in}$. The factor of two in $p(\mathbf{x})dA/2$ is due to the symmetry of Poisson fields in $z < 0$ and $z > 0$.

### 3.1  Augment the data with additional dimension

We wish to generate samples $\mathbf{x} \in \mathbb{R}^N$ from a distribution $p(\mathbf{x})$ supported on a bounded region. We may set the source $\rho(\mathbf{x}) = p(\mathbf{x}) \in \mathcal{C}^0$ [4] and compute the resulting gradient field $\mathbf{E}(\mathbf{x})$ from Eq. (3). Since $-\mathbf{E}(\mathbf{x})$ points towards sources, the backward ODE $d\mathbf{x}/dt = -\mathbf{E}(\mathbf{x})$ will take samples close to the sources. One may naively hope that the backward ODE is a generative model that recovers $p(\mathbf{x})$. Unfortunately, the backward ODE has the problem of mode collapse. We illustrate this phenomenon with a 2D uniform disk. The reverse Poisson field $-\mathbf{E}(\mathbf{x})$ on the 2D $(x, y)$-plane points towards the center of the disk $O$ (Fig. 2(a) left), so all particle trajectories (blue lines) will eventually hit $O$. If we instead add an additional dimension $z$ (Fig. 2(a) right), particles can hit different points on the disk and faithfully recover the data distribution.

Consequently, instead of solving the Poisson equation $\nabla^2 \varphi(\mathbf{x}) = -p(\mathbf{x})$ in the original data space, we solve the Poisson equation in an augmented space $\tilde{\mathbf{x}} = (\mathbf{x}, z) \in \mathbb{R}^{N+1}$ with an additional variable $z \in \mathbb{R}$. We augment the training data $\tilde{\mathbf{x}}$ in the new space by setting $z = 0$ such that $\tilde{\mathbf{x}} = (\mathbf{x}, 0)$. As a consequence, the data distribution in the augmented space is $\tilde{p}(\tilde{\mathbf{x}}) = p(\mathbf{x})\delta(z)$, where $\delta$ is the Dirac delta function. By Eq. (3), the Poisson field by solving the new Poisson equation $\nabla^2 \varphi(\tilde{\mathbf{x}}) = -\tilde{p}(\tilde{\mathbf{x}})$ has an analytical form:

$$\forall \tilde{\mathbf{x}} \in \mathbb{R}^{N+1}, \mathbf{E}(\tilde{\mathbf{x}}) = -\nabla\varphi(\tilde{\mathbf{x}}) = \frac{1}{S_N(1)} \int \frac{\tilde{\mathbf{x}} - \tilde{\mathbf{y}}}{\|\tilde{\mathbf{x}} - \tilde{\mathbf{y}}\|^{N+1}} \tilde{p}(\tilde{\mathbf{y}})d\tilde{\mathbf{y}} \tag{4}$$

The associated forward/backward ODEs of the Poisson field are $d\tilde{\mathbf{x}}/dt = \mathbf{E}(\tilde{\mathbf{x}}), d\tilde{\mathbf{x}}/dt = -\mathbf{E}(\tilde{\mathbf{x}})$. Intuitively, theses ODEs uniquely define trajectories of particles between the $z = 0$ hyperplane and an enclosing hemisphere (cf. Fig. 1(a)). In the following theorem, we show that the backward ODE defines a transformation between the uniform distribution on an infinite hemisphere and the data distribution $\tilde{p}(\tilde{\mathbf{x}})$ in the $z = 0$ plane. We present the formal proof to Appendix A, illustrated by Fig. 2(b). The proof is based on the idea that when the radius of hemisphere $r \to \infty$, the data distribution $\tilde{p}(\tilde{\mathbf{x}})$ can be effectively viewed as a delta distribution at origin. Consequently, the Poisson field points in the radial direction at $r \to \infty$, perpendicular to $S_N^+(r)$ (Green arrows in Fig. 2(b)).

**Theorem 1.** *Suppose particles are sampled from a uniform distribution on the upper ($z > 0$) half of the sphere of radius $r$ and evolved by the backward ODE $\frac{d\tilde{\mathbf{x}}}{dt} = -\mathbf{E}(\tilde{\mathbf{x}})$ until they reach the $z = 0$ hyperplane, where the Poisson field $\mathbf{E}(\tilde{\mathbf{x}})$ is generated by the source $\tilde{p}(\tilde{\mathbf{x}})$. In the $r \to \infty$ limit, under some mild conditions detailed in Appendix A, this process generates a particle distribution $\tilde{p}(\tilde{\mathbf{x}})$, i.e., a distribution $p(\mathbf{x})$ in the $z = 0$ hyperplane.*

*Proof sketch.* Suppose the flux of the backward ODE connects a solid angle $d\Omega$ (on $S_N^+(r)$) with an area $dA$ (on $\text{supp}(\tilde{p}(\tilde{\mathbf{x}}))$). According to Gauss's law, the outflow flux $d\Phi_{out} = d\Omega/S_N(1)$ on the

---

[4]A probability distribution $p(\mathbf{x})$ is a special case of "charge density" $\rho(x)$ because $p(\mathbf{x})$ need to be non-negative and integrates to unity. Here we focus on applications to probability distribution of data, which is the objective to be modeled in generative modeling.

hemisphere (Green arrows in Fig. 2(b)) equals the inflow flux $d\Phi_{in} = p(\mathbf{x})dA/2$ on $\mathrm{supp}(\tilde{p}(\tilde{\mathbf{x}}))$ (Red arrows in Fig. 2(b)). $d\Phi_{in} = d\Phi_{out}$ gives $d\Omega/dA = p(\mathbf{x})S_N(1)/2 \propto p(\mathbf{x})$. Together, by change-of-variable, we conclude that the final distribution in the $z = 0$ hyperplane is $p(\mathbf{x})$. $\qquad\square$

The theorem states that starting from an infinite hemisphere, one can recover the data distribution $\tilde{p}$ by following the inverse Poisson field $-\mathbf{E}(\tilde{\mathbf{x}})$. We defer the formal proof and technical assumptions of the theorem to Appendix A. The property allows generative modeling by following the Poisson flow of $\nabla^2 \varphi(\tilde{\mathbf{x}}) = -\tilde{p}(\tilde{\mathbf{x}})$.

## 3.2 Learning the normalized Poisson Field

Given a set of training data $\mathcal{D} = \{\mathbf{x}_i\}_{i=1}^n$ i.i.d sampled from the data distribution $p(\mathbf{x})$, we define the empirical version of the Poisson field (Eq. (4)) as follows:

$$\hat{\mathbf{E}}(\tilde{\mathbf{x}}) = c(\tilde{\mathbf{x}}) \sum_{i=1}^n \frac{\tilde{\mathbf{x}} - \tilde{\mathbf{x}}_i}{\|\tilde{\mathbf{x}} - \tilde{\mathbf{x}}_i\|^{N+1}}$$

where the gradient field is calculated on $n$ *augmented* datapoints $\{\tilde{\mathbf{x}}_i = (\mathbf{x}_i, 0)\}_{i=1}^n$, and $c(\tilde{\mathbf{x}}) = 1/\sum_{i=1}^n \frac{1}{\|\tilde{\mathbf{x}} - \tilde{\mathbf{x}}_i\|^{N+1}}$ is the multiplier for numerical stability. We further normalize the field to resolve the variations in the magnitude of the norm $\| \hat{\mathbf{E}}(\tilde{\mathbf{x}}) \|_2$, and fit the neural network to the more amenable negative normalized field $\mathbf{v}(\tilde{\mathbf{x}}) = -\sqrt{N}\hat{\mathbf{E}}(\tilde{\mathbf{x}})/\| \hat{\mathbf{E}}(\tilde{\mathbf{x}}) \|_2$. The Poisson field is rescalable (cf. Section 2) and thus trajectories of its forward/backward ODEs are invariant under normalization. We denote the empirical field calculated on batch data $\mathcal{B}$ by $\hat{\mathbf{E}}_{\mathcal{B}}$ and the negative normalized field as $\mathbf{v}_{\mathcal{B}}(\tilde{\mathbf{x}}) = -\sqrt{N}\hat{\mathbf{E}}_{\mathcal{B}}(\tilde{\mathbf{x}})/\| \hat{\mathbf{E}}_{\mathcal{B}}(\tilde{\mathbf{x}}) \|_2$.

Similar to the scored-based models, we sample points inside the hemisphere by perturbing the augmented training data. Given a training point $\mathbf{x} \in \mathcal{D}$, we add noise to its augmented version $\{\tilde{\mathbf{x}}_i = (\mathbf{x}_i, 0)\}_{i=1}^n$ to construct the perturbed point $(\mathbf{y}, z)$:

$$\mathbf{y} = \mathbf{x} + \| \epsilon_{\mathbf{x}} \| (1 + \tau)^m \mathbf{u}, \quad z = |\epsilon_z|(1 + \tau)^m \tag{5}$$

where $\epsilon = (\epsilon_{\mathbf{x}}, \epsilon_z) \sim \mathcal{N}(0, \sigma^2 I_{N+1 \times N+1})$, $\mathbf{u} \sim \mathcal{U}(S_{N-1}(1))$ and $m \sim \mathcal{U}[0, M]$. The upper limit $M$, standard deviation $\sigma$ and $\tau$ are hyper-parameters. With fixed $\epsilon$ and $\mathbf{u}$, the added noise increases exponentially with $m$. The rationale behind the design is that points farther away from the data support play a less important role in generative modeling, sharing a similar spirit with the choice of noisy scales in score-based models [32, 33].

In practice, we sample the points by perturbing a mini-batch data $\mathcal{B} = \{\mathbf{x}_i\}_{i=1}^{|\mathcal{B}|}$ in each iteration. We uniformly sample the power $m$ in $[0, M]$ for each datapoint. We select a large $M$ (typically around 300) to ensure the perturbed points can reach a large enough hemisphere. We use a larger batch $\mathcal{B}_L$ for the estimation of normalized field since the empirical normalized field is biased, which empirically gives better results. Denoting the set of perturbed points as $\{\tilde{\mathbf{y}}_i\}_{i=1}^{|\mathcal{B}|}$, we train the neural network $f_\theta$ on these points to estimate the negative normalized field by minimizing the following loss:

$$\mathcal{L}(\theta) = \frac{1}{|\mathcal{B}|} \sum_{i=1}^{|\mathcal{B}|} \| f_\theta(\tilde{\mathbf{y}}_i) - \mathbf{v}_{\mathcal{B}_L}(\tilde{\mathbf{y}}_i) \|_2^2$$

We summarize the training process in Algorithm 1. In practice, we add a small constant $\gamma$ to the denominator of the normalized field to overcome the numerical issue when $\exists i, \|\tilde{\mathbf{x}} - \tilde{\mathbf{x}}_i\| \approx 0$.

## 3.3 Backward ODE anchored by the additional dimension

After estimating the normalized field $\mathbf{v}$, we can sample from the data distribution by the backward ODE $d\tilde{\mathbf{x}} = \mathbf{v}(\tilde{\mathbf{x}})dt$. Nevertheless, the boundary condition of the above ODE is unclear: the starting and terminal time $t$ of the ODE are both unknown. To remedy the issue, we propose an equivalent backward ODE in which $\mathbf{x}$ evolves with the augmented variable $z$:

$$d(\mathbf{x}, z) = (\frac{d\mathbf{x}}{dt}\frac{dt}{dz}dz, dz) = (\mathbf{v}(\tilde{\mathbf{x}})_{\mathbf{x}}\mathbf{v}(\tilde{\mathbf{x}})_z^{-1}, 1)dz$$

---

**Algorithm 1** Learning the normalized Poisson Field

---

**Input:** Training iteration $T$, Initial model $f_\theta$, dataset $\mathcal{D}$, constant $\gamma$, learning rate $\eta$.
**for** $t = 1 \dots T$ **do**
    Sample a large batch $\mathcal{B}_L$ from $\mathcal{D}$ and subsample a batch of datapoints $\mathcal{B} = \{\mathbf{x}_i\}_{i=1}^{|\mathcal{B}|}$ from $\mathcal{B}_L$
    Simulate the ODE: $\{\tilde{\mathbf{y}}_i = \mathrm{perturb}(\mathbf{x}_i)\}_{i=1}^{|\mathcal{B}|}$
    Calculate the normalized field by $\mathcal{B}_L$: $\mathbf{v}_{\mathcal{B}_L}(\tilde{\mathbf{y}}_i) = -\sqrt{N}\hat{\mathbf{E}}_{\mathcal{B}_L}(\tilde{\mathbf{y}}_i)/(\|\hat{\mathbf{E}}_{\mathcal{B}_L}(\tilde{\mathbf{y}}_i)\|_2 + \gamma), \forall i$
    Calculate the loss: $\mathcal{L}(\theta) = \frac{1}{|\mathcal{B}|}\sum_{i=1}^{|\mathcal{B}|}\| f_\theta(\tilde{\mathbf{y}}_i) - \mathbf{v}_{\mathcal{B}_L}(\tilde{\mathbf{y}}_i)\|_2^2$
    Update the model parameter: $\theta = \theta - \eta\nabla\mathcal{L}(\theta)$
**end for**
**return** $f_\theta$

---

---

**Algorithm 2** perturb($\mathbf{x}$)

---

Sample the power $m \sim \mathcal{U}[0, M]$
Sample the initial noise $(\epsilon_\mathbf{x}, \epsilon_z) \sim \mathcal{N}(0, \sigma^2 I_{(N+1)\times(N+1)})$
Uniformly sample the vector from the unit ball $\mathbf{u} \sim \mathcal{U}(S_N(1))$
Construct training point $\mathbf{y} = \mathbf{x} + \|\epsilon_\mathbf{x}\|(1+\tau)^m\mathbf{u}, z = |\epsilon_z|(1+\tau)^m$
**return** $\tilde{\mathbf{y}} = (\mathbf{y}, z)$

---

where $\mathbf{v}(\tilde{\mathbf{x}})_\mathbf{x}, \mathbf{v}(\tilde{\mathbf{x}})_z$ are the corresponding components of $\mathbf{x}, z$ in vector $\mathbf{v}(\tilde{\mathbf{x}})$. In the new ODE, we replace the time variable $t$ with the physically meaningful variable $z$, permitting explicit starting and terminal conditions: when $z = 0$, we arrive at the data distribution and we can freely choose a large $z_{\max}$ as the starting point in the backward ODE. The backward ODE is compatible with general-purpose ODE solvers, *e.g.*, RK45 method [23] and forward Euler method. The popular black-box ODE solvers, such as the one in Scipy library [37], typically use a common starting time for the same batch of samples. Since the distribution on the $z = z_{\max}$ hyperplane is no longer uniform, we derive the prior distribution by radially projecting uniform distribution on the hemisphere with radius $r = z_{\max}$ to the $z = z_{\max}$ hyperplane:

$$p_{\mathrm{prior}}(\mathbf{x}) = \frac{2z_{\max}^{N+1}}{S_N(z_{\max})(\|\mathbf{x}\|_2^2 + z_{\max}^2)^{\frac{N+1}{2}}} = \frac{2z_{\max}}{S_N(1)(\|\mathbf{x}\|_2^2 + z_{\max}^2)^{\frac{N+1}{2}}}$$

where $S_N(r)$ is the surface area of $N$-sphere with radius $r$. The reason behind the radial projection is that the Poisson field points in the radial direction at $r \to \infty$. The new backward ODE also defines a bijective transformation between $p_{\mathrm{prior}}(\mathbf{x})$ on the infinite hyperplane ($z_{\max} \to \infty$) and the data distribution $\tilde{p}(\tilde{\mathbf{x}})$, analogous to Theorem 1. In order to sample from $p_{\mathrm{prior}}(\mathbf{x})$, it is suffice to sample the norm (radius) from the distribution: $p_{\mathrm{radius}}(\|\mathbf{x}\|_2) \propto \|\mathbf{x}\|_2^{N-1}/(\|\mathbf{x}\|_2^2 + z_{\max}^2)^{\frac{N+1}{2}}$ and then uniformly sample its angle. We provide detailed derivations and practical sampling procedure in Appendix A.4. We further achieve exponential decay on the $z$ dimension by introducing a new variable $t'$:

$$[\text{Backward ODE}] \quad d(\mathbf{x}, z) = (\mathbf{v}(\tilde{\mathbf{x}})_\mathbf{x}\mathbf{v}(\tilde{\mathbf{x}})_z^{-1}z, z)dt' \tag{6}$$

The $z$ component in the backward ODE, *i.e.*, $dz = zdt'$, can be solved by $z = e^{t'}$. Since $z$ reaches zero as $t' \to -\infty$, we instead choose a tiny positive number $z_{\min}$ as the terminal condition. The corresponding starting/terminal time of the variable $t'$ are $\log z_{\max}/\log z_{\min}$ respectively. Empirically, this simple change of variable leads to 2× faster sampling with almost no harm to the sample quality. In addition, we substitue the predicted $\mathbf{v}(\tilde{\mathbf{x}})_z$ with a more accurate one when $z$ is small (Appendix B.2.3). We defer more details of the simulation of backward ODE to Appendix B.2.

## 4 Generative Modeling via the Backward ODE

In this section, we demonstrate the effectiveness of the backward ODE associated with PFGM on image generation tasks. In Section 4.1, we show that PFGM achieves currently best in class performance in the normalizing flow family. In comparison to the existing state-of-the-art SDE or MCMC approaches, PFGM exhibits 10× or 20× acceleration while maintaining competitive or

Table 1: CIFAR-10 sample quality (FID, Inception) and number of function evaluation (NFE).

| | Invertible? | Inception ↑ | FID ↓ | NFE ↓ |
|---|---|---|---|---|
| PixelCNN [36] | ✗ | 4.60 | 65.9 | 1024 |
| IGEBM [8] | ✗ | 6.02 | 40.6 | 60 |
| ViTGAN [24] | ✗ | 9.30 | 6.66 | 1 |
| StyleGAN2-ADA [17] | ✗ | 9.83 | 2.92 | 1 |
| StyleGAN2-ADA (cond.) [17] | ✗ | 10.14 | 2.42 | 1 |
| NCSN [31] | ✗ | 8.87 | 25.32 | 1001 |
| NCSNv2 [32] | ✗ | 8.40 | 10.87 | 1161 |
| DDPM [16] | ✗ | 9.46 | 3.17 | 1000 |
| NCSN++ VE-SDE [33] | ✗ | 9.83 | 2.38 | 2000 |
| NCSN++ deep VE-SDE [33] | ✗ | 9.89 | 2.20 | 2000 |
| Glow [19] | ✓ | 3.92 | 48.9 | 1 |
| DDIM, T=50 [30] | ✓ | - | 4.67 | 50 |
| DDIM, T=100 [30] | ✓ | - | 4.16 | 100 |
| NCSN++ VE-ODE [33] | ✓ | 9.34 | 5.29 | 194 |
| NCSN++ deep VE-ODE [33] | ✓ | 9.17 | 7.66 | 194 |
| *DDPM++ backbone* | | | | |
| VP-SDE [33] | ✗ | 9.58 | 2.55 | 1000 |
| sub-VP-SDE [33] | ✗ | 9.56 | 2.61 | 1000 |
| VP-ODE [33] | ✓ | 9.46 | 2.97 | 134 |
| sub-VP-ODE [33] | ✓ | 9.30 | 3.16 | 146 |
| PFGM (ours) | ✓ | **9.65** | **2.48** | **104** |
| *DDPM++ deep backbone* | | | | |
| VP-SDE [33] | ✗ | 9.68 | 2.41 | 1000 |
| sub-VP-SDE [33] | ✗ | 9.57 | 2.41 | 1000 |
| VP-ODE [33] | ✓ | 9.47 | 2.86 | 134 |
| sub-VP-ODE [33] | ✓ | 9.40 | 3.05 | 146 |
| PFGM (ours) | ✓ | **9.68** | **2.35** | **110** |

higher generation quality. Meanwhile, unlike existing ODE baselines that heavily rely on corrector to generate decent samples on weaker architectures, PFGM exhibits greater stability against error (Section 4.2). Finally, we show that PFGM is robust to the step size in the Euler method (Section 4.3), and its associated ODE allows for likelihood evaluation and image manipulation by editing the latent space (Section 4.4).

## 4.1 Efficient image generation by PFGM

**Setup**   For image generation tasks, we consider the CIFAR-10 [22], CelebA $64 \times 64$ [38] and LSUN bedroom $256 \times 256$ [39]. Following [32], we first center-crop the CelebA images and then resize them to $64 \times 64$. We choose $M = 291$ (CIFAR-10 and CelebA)/356 (LSUN bedroom), $\sigma = 0.01$ and $\tau = 0.03$ for the perturbation Algorithm 2, and $z_{min} = 1e - 3$, $z_{max} = 40$ (CIFAR-10)/60 (CelebA $64^2$)/100 (LSUN bedroom) for the backward ODE. We further clip the norms of initial samples into $(0, 3000)$ for CIFAR-10, $(0, 6000)$ for CelebA $64^2$ and $(0, 30000)$ for LSUN bedroom. We adopt the DDPM++ and DDPM++ deep architectures [33] as our backbones. We add the scalar $z$ (resp. predicted direction on $z$) as input (resp. output) to accommodate the additional dimension. We take the same set of hyper-parameters, such as batch size, learning rate and training iterations from [33]. We provide more training details in Appendix B.1, and discuss how to set these hyper-parameters for general datasets in B.1.1 and B.2.1.

**Baselines**   We compare PFGM to modern autoregressive model [36], GAN [17, 24], normalizing flow [19] and EBM [8]. We also compare with variants of score-based models such as DDIM [30] and current state-of-the-art SDE/ODE methods [33]. We denote the methods that use forward-time SDEs in [33] such as Variance Exploding (**VE**) SDE/Variance Preserving (**VP**) SDE/ sub-Variance Preserving (**sub-VP**), and the corresponding backward SDE/ODE, as **A-B**, where A ∈ {VE, VP, sub-VP} and B ∈ {SDE, ODE}. We follow the model selection protocol in [33], which selects the checkpoint with the smallest FID score over the course of training every 50k iterations.

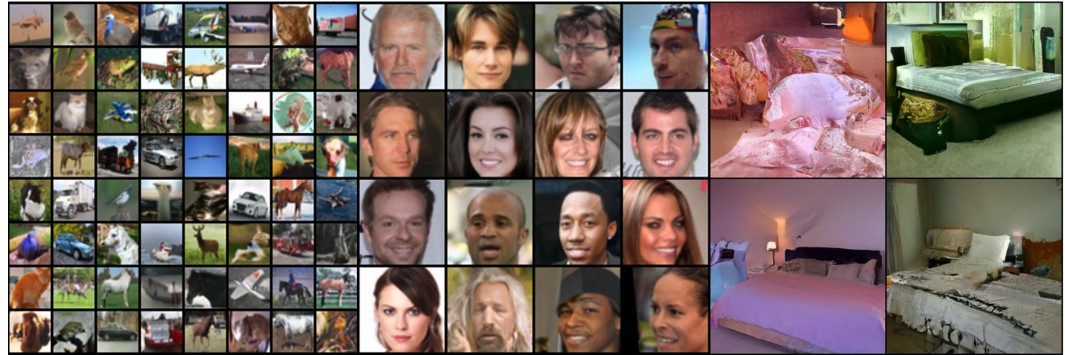

Figure 3: Uncurated samples on datasets of increasing resolution. From left to right: CIFAR-10 $32 \times 32$, CelebA $64 \times 64$ and LSUN bedroom $256 \times 256$.

**Numerical Solvers**   The backward ODE (Eq. (6)) is compatible with any general purpose ODE solver. In our experiments, the default solver of ODEs is the black box solver in the Scipy library [37] with the RK45 [7] method (**RK45**), unless otherwise specified. For VE/VP/subVP-SDEs, we use the predictor-corrector (**PC**) sampler introduced in [33]. For VP/sub-VP-SDEs, we apply the predictor-only sampler, because its performance is on par with the PC sampler while requiring half computation.

**Results**   For quantitative evaluation on CIFAR-10, we report the Inception [29] (higher is better) and FID [13] scores (lower is better) in Table 1. We also include our preliminary experimental results on a weaker architecture NCSNv2 [32] in Appendix D.2. We measure the inference speed by the average NFE (number of function evaluation). We also explicitly indicate which methods belong to the invertible flow family.

Our main findings are: **(1) PFGM achieves the best Inception scores and FID scores among the normalizing flow models.** Specifically, PFGM obtains an Inception score of 9.68 and a FID score of 2.48 using the DDPM++ deep architecture. To our best knowledge, these are the highest FID and Inception scores by flow models on CIFAR-10. **(2) PFGM achieves a** $10\times \sim 20\times$ **faster inference speed than the SDE methods using the similar architectures, while retaining comparable sample quality.** As shown in Table 1, PFGM requires NFEs of 110 whereas the SDE methods typically use $1000 \sim 2000$ inference steps. PFGM outperforms all the baselines on DDPM++ in all metrics. In addition, PFGM generally samples faster than other ODE baselines with the same RK45 solver. **(3) The backward ODE in PFGM is compatible with architectures with varying capacities.** PFGM consistently outperforms other ODE baselines on DDPM++ (Table 1) or NCSNv2 (Appendix D.2) backbones. **(4) PFGM shows scalability to higher resolution datasets.** In Appendix D.1, we show that PFGM are capable of scale-up to LSUN bedroom $256 \times 256$. In particular, PFGM has comparable performance with VE-SDE with $15\times$ fewer NFE.

In Fig. 3, we visualize the uncurated samples from PFGM on CIFAR-10, CelebA $64 \times 64$ and LSUN bedroom $256 \times 256$. We provides more samples in Appendix E.

## 4.2   Failure of VE/VP-ODEs on NCSNv2 architecture

In our preliminary experiments on NCSNv2 architectures, we empirically observe that the VE/VP-ODEs have FID scores greater than 90 on CIFAR-10. In particular, VE/VP-ODEs can only generate decent samples when applying the Langevin dynamics corrector, and even then, their performances are still inferior to PFGM (Table 9, Table 10). The poor performance on NCSNv2 stands in striking contrast to their high sample quality on NCSN++/DDPM++ in [33]. **It indicates that the VE/VP-ODEs are**

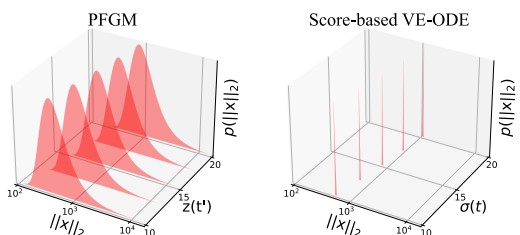

Figure 4: Sample norm distributions with varying time variables ($\sigma$ for VE-ODE and $z$ for PFGM)

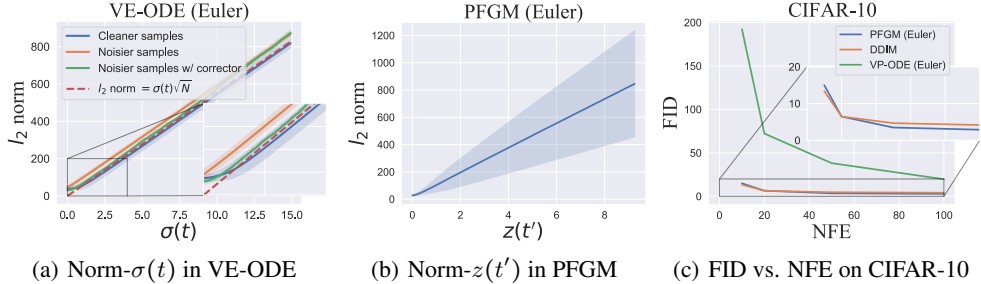

(a) Norm-$\sigma(t)$ in VE-ODE     (b) Norm-$z(t')$ in PFGM     (c) FID vs. NFE on CIFAR-10

Figure 5: **(a)** Norm-$\sigma(t)$ relation during the backward sampling of VE-ODE (Euler). **(b)** Norm-$z(t')$ relation during the backward sampling of PFGM (Euler). The shaded areas mean the standard deviation of norms. **(c)** Number of steps versus FID score.

**more susceptible to estimation errors than PFGM.** We hypothesize that the strong norm-$\sigma$ correlation seen during the training of score-based models causes the problem.

For score-based models, the $l_2$ norms of perturbed training samples and the standard deviations $\sigma(t)$ of Gaussian noises have strong correlation, *e.g.*, $l_2$ norm $\approx \sigma(t)\sqrt{N}$ for large $\sigma(t)$ in VE [33]. In contrast, as shown in Fig. 4, PFGM allocates high mass across a wide spectrum of the training sample norms. During sampling, VE/VP-ODEs could break down when the trajectories of backward ODEs deviate from the norm-$\sigma(t)$ relation to which most training samples pertain. The weaker NCSNv2 backbone incurs larger errors and thus leads to their failure. The PFGM is more resistant to estimate errors because of the greater range of training sample norms.

To further verify the hypothesis above, we split a batch of VE-ODE samples into cleaner and noisier samples according to visual quality (Fig. 8(a)). In Fig. 5(a), we investigate the relation for cleaner and noisier samples during the forward Euler simulation of VE-ODE when $\sigma(t) < 15$. We can see that the trajectory of cleaner samples stays close to the norm-$\sigma(t)$ relation (the red dash line), whereas that of the noisier samples diverges from the relation. The Langevin dynamics corrector changes the trajectory of noisier samples to align with the relation. Fig. 5(b) further shows that the anchored variable $z(t')$ and the norms in the backward ODE of PFGM are not strongly correlated, giving rise to the robustness against the imprecise estimation on NCSNv2. We defer more details to Appendix C.

### 4.3   Effects of step size in the forward Euler method

In order to accelerate the inference speed of ODEs, we can increase the step size (decrease the NFEs) in numerical solvers such as the forward Euler method. It also enables the trade-off between sample quality and computational efficiency in real-world deployment. We study the effects of increasing step size on PFGM, VP-ODE and DDIM [30] using the forward Euler method, with a varying NFE ranging from 10 to 100.

In Fig. 5(c), we report the sample quality measured by FID scores on CIFAR-10. As expected, all the methods have higher FID scores when decreasing the NFE. We observe that the sample quality of PFGM degrades gracefully as we decrease the NFE. Our method shows significantly better robustness to step sizes than the VP-ODE, especially when only taking a few Euler steps. In addition, PFGM obtains better FID scores than DDIM on most NFEs except for 10 where PFGM is marginally worse. This suggests that the PFGM is a promising method for accommodating instantaneous resource availability, as high-quality samples can be generated in limited steps.

### 4.4   Utilities of ODE: likelihood evaluation and latent representation

Similar to the family of discrete normalizing flows [6, 19, 14] and continuous probability flow [33], the forward ODE in PFGM defines an invertible mapping between the data space and latent space with a known prior. Formally, we define the invertible forward $\mathcal{M}$ mapping by integrating the corresponding forward ODE $d(\mathbf{x}, z) = (\mathbf{v}(\tilde{\mathbf{x}})_\mathbf{x}\mathbf{v}(\tilde{\mathbf{x}})_z^{-1}z, z)dt'$ of Eq. (6):

$$\mathbf{x}(\log z_{\max}) = \mathcal{M}(\mathbf{x}(\log z_{\min})) \equiv \mathbf{x}(\log z_{\min}) + \int_{\log z_{\min}}^{\log z_{\max}} \mathbf{v}(\mathbf{x}(t'))_\mathbf{x}\mathbf{v}(\tilde{\mathbf{x}}(t'))_z^{-1}e^{t'}dt'$$

where $\log z_{\min}/\log z_{\max}$ are the starting/terminal time in the forward ODE. The forward mapping transfers the data distribution to the prior distribution $p_{\text{prior}}$ on the $z = z_{\max}$ hyperplane (cf. Section 3.3): $p_{\text{prior}}(\mathbf{x}(\log z_{\max})) = \mathcal{M}(p(\mathbf{x}(\log z_{\min})))$. The invertibility enables likelihood evaluation and creates a meaningful latent space on the $z = z_{\max}$ hyperplane. In addition, we can adapt to the computational constraints by adjusting the step size or the precision in numerical ODE solvers.

**Likelihood evaluation** We evaluate the data likelihood by the instantaneous change-of-variable formula [4, 33]. In Table 2, we report the bits/dim on the uniformly de-quantized CIFAR-10 test set and compare with existing baselines that use the same setup. We observe that PFGM achieves better likelihoods than discrete normalizing flow models, even without maximum likelihood training. Among the continuous flow models, sub-VP-ODE shows the lowest bits/dim, although its sample quality is worse than VP-ODE and PFGM (Table 1). The exploration of the seeming trade-off between likelihood and sample quality is left for future works.

Table 2: Bits/dim on CIFAR-10

|  | bits/dim $\downarrow$ |
| --- | --- |
| RealNVP [6] | 3.49 |
| Glow [19] | 3.35 |
| Residual Flow [3] | 3.28 |
| Flow++ [14] | 3.29 |
| DDPM ($L$) [16] | $\leq 3.70^{*}$ |
| *DDPM++ backbone* |  |
| VP-ODE [33] | 3.20 |
| sub-VP-ODE [33] | **3.02** |
| PFGM (ours) | 3.19 |

**Latent representation** Since the samples are uniquely identifiable by their latents via the invertible mapping $\mathcal{M}$, PFGM further supports image manipulation using its latent representation on the $z = z_{\max}$ hyperplane. We include the results of image interpolation and the temperature scaling [6, 19, 33] to Appendix D.4 and Appendix D.5. For interpolation, it shows that we can travel along the latent space to obtain perceptually consistent interpolations between CelebA images.

## 5 Conclusion

We present a new deep generative model by solving the Poisson equation whose source term is the data distribution. We estimate the normalized gradient field of the solution in an augmented space with an additional dimension. For sampling, we devise a backward ODE that exponential decays on the physically meaningful additional dimension. Empirically, our approach has currently best performance over other normalizing flow baselines, and achieving 10× to 20× acceleration over the stochastic methods. Our backward ODE shows greater stability against errors than popular ODE-based methods, and enables efficient adaptive sampling. We further demonstrate the utilities of the forward ODE on likelihood evaluation and image interpolation. Future directions include improving the normalization of Poisson fields. More principled approaches can be used to get around the divergent near-field behavior. For example, we may exploit renormalization, a useful tool in physics, to make the Poisson field well-behaved in near fields.

## Acknowledgements

We are grateful to Shangyuan Tong, Timur Garipov and Yang Song for helpful discussion. We would like to thank Octavian Ganea and Wengong Jin for reviewing an early draft of this paper. YX and TJ acknowledge support from MIT-DSTA Singapore collaboration, from NSF Expeditions grant (award 1918839) "Understanding the World Through Code", and from MIT-IBM Grand Challenge project. ZL and MT would like to thank the Center for Brains, Minds, and Machines (CBMM) for hospitality. ZL and MT are supported by The Casey and Family Foundation, the Foundational Questions Institute, the Rothberg Family Fund for Cognitive Science and IAIFI through NSF grant PHY-2019786.

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
