# Appendix

## A  Proofs

### A.1  Formal Proof of Theorem 1

Before proceeding to Theorem 1, we show a technical lemma that guarantees the existence-uniqueness of the solution to the Poisson equation, under some mild conditions.

**Lemma 1.** *Given $\Omega = \mathbb{R}^N, N \geq 3$, assume that the source function $\rho \in \mathcal{C}^0(\Omega)$, and $\rho$ has a compact support. Then the the Poisson equation $\nabla^2 \varphi(\mathbf{x}) = -\rho(\mathbf{x})$ on $\Omega$ with zero boundary condition at infinity ($\lim_{\|\mathbf{x}\|_2 \to \infty} \varphi(\mathbf{x}) = 0$) has a unique solution $\varphi(\mathbf{x}) \in \mathcal{C}^2(\Omega)$ up to a constant.*

*Proof.* For the existence of the solution, one can verify that the analytical construction using the extension of Green's function in $N \geq 3$ dimensional space (Lemma 4), *i.e.*, $\varphi(\mathbf{x}) = \int G(\mathbf{x}, \mathbf{y}) \rho(\mathbf{y}) d\mathbf{y}, G(\mathbf{x}, \mathbf{y}) = \frac{1}{(N-2)S_{N-1}(1)} \frac{1}{\|\mathbf{x}-\mathbf{y}\|^{N-2}}$, is one possible solution to the Poisson equation $\nabla^2 \varphi(\mathbf{x}) = -\rho(\mathbf{x})$. Since $\rho \in \mathcal{C}^0(\Omega)$ and $\nabla^2 \varphi(\mathbf{x}) = -\rho(\mathbf{x})$, we conclude that $\varphi(\mathbf{x}) \in \mathcal{C}^2(\Omega)$.

The proof idea of the uniqueness is similar to the uniqueness theorems in electrostatics. Suppose we have two different solutions $\varphi_1, \varphi_2 \in \mathcal{C}^2$ which satisfy

$$\nabla^2 \varphi_1(\mathbf{x}) = -\rho(\mathbf{x}), \nabla^2 \varphi_2(\mathbf{x}) = -\rho(\mathbf{x}). \tag{7}$$

We define $\tilde{\varphi}(\mathbf{x}) \equiv \varphi_2(\mathbf{x}) - \varphi_1(\mathbf{x})$. Subtracting the above two equations gives

$$\nabla^2 \tilde{\varphi}(\mathbf{x}) = 0, \forall \mathbf{x} \in \Omega. \tag{8}$$

By the vector differential identity we have

$$\tilde{\varphi}(\mathbf{x}) \nabla^2 \tilde{\varphi}(\mathbf{x}) = \nabla \cdot (\tilde{\varphi}(\mathbf{x}) \nabla \tilde{\varphi}(\mathbf{x})) - \nabla \tilde{\varphi}(\mathbf{x}) \cdot \nabla \tilde{\varphi}(\mathbf{x}), \tag{9}$$

By the divergence theorem we have

$$\int_\Omega \nabla \cdot (\tilde{\varphi}(\mathbf{x}) \nabla \tilde{\varphi}(\mathbf{x})) d^N \mathbf{x} = \oiint_{\partial \Omega} \tilde{\varphi}(\mathbf{x}) \nabla \tilde{\varphi}(\mathbf{x}) \cdot d^{N-1} \mathbf{S} = 0, \tag{10}$$

where $d^{N-1} \mathbf{S}$ denotes an $N-1$ dimensional surface element at infinity, and the second equation holds due to zero boundary condition at infinity. Combining Eq. (8)(9)(10), we have

$$\int_\Omega \nabla \cdot (\tilde{\varphi}(\mathbf{x}) \nabla \tilde{\varphi}(\mathbf{x})) d^N \mathbf{x} = \int_\Omega \|\nabla \tilde{\varphi}(\mathbf{x})\|^2 d^N \mathbf{x} = 0, \tag{11}$$

since this is an integral of a positive quantity, we must have $\nabla \tilde{\varphi}(\mathbf{x}) = \mathbf{0}$, or $\tilde{\varphi}(\mathbf{x}) = c, \forall \mathbf{x} \in \Omega$. This means $\varphi_1$ and $\varphi_2$ differ at most by a constant, but a constant does not affect gradients, so $\nabla \varphi_1(\mathbf{x}) = \nabla \varphi_2(\mathbf{x})$. $\qquad \square$

In our method section (Section 3.1), we augmented the original $N$-dimensional data with an extra dimension. The new data distribution in the augmented space is $\tilde{p}(\tilde{\mathbf{x}}) = p(\mathbf{x})\delta(z)$, where $\delta$ is the Dirac delta function. The support of the data distribution is in the $z = 0$ hyperplane. In the following lemma, we show the existence and uniqueness of the solution to $\nabla^2 \varphi(\tilde{\mathbf{x}}) = -\tilde{p}(\tilde{\mathbf{x}})$ outside the data support.

**Lemma 2.** *Assume the support of the data distribution in the augmented space ($supp(\tilde{p}(\tilde{\mathbf{x}}))$) is a compact set on the $z = 0$ hyperplane, $p(\mathbf{x}) \in \mathcal{C}^0$ and $N \geq 3$. The Poisson equation $\nabla^2 \varphi(\tilde{\mathbf{x}}) = -\tilde{p}(\tilde{\mathbf{x}})$ with zero boundary condition at infinity ($\lim_{\|\mathbf{x}\|_2 \to \infty} \varphi(\tilde{\mathbf{x}}) = 0$) has a unique solution $\varphi(\tilde{\mathbf{x}}) \in \mathcal{C}^2$ for $\tilde{x} \in \mathbb{R}^{N+1} \setminus supp(\tilde{p}(\tilde{\mathbf{x}}))$, up to a constant.*

*Proof.* Similar to the proof in Lemma 1, one can easily verify that the analytical construction using Green's method, *i.e.*, $\varphi(\tilde{\mathbf{x}}) = \int G(\tilde{\mathbf{x}}, \tilde{\mathbf{y}}) \tilde{p}(\tilde{\mathbf{x}}) d\tilde{\mathbf{y}}, G(\tilde{\mathbf{x}}, \tilde{\mathbf{y}}) = \frac{1}{(N-1)S_N(1)} \frac{1}{\|\tilde{\mathbf{x}}-\tilde{\mathbf{y}}\|^{N-1}}$, is one possible solution to the Poisson equation $\nabla^2 \varphi(\tilde{\mathbf{x}}) = -\tilde{p}(\tilde{\mathbf{x}})$. Since $\tilde{p}(\tilde{\mathbf{x}}) = 0$ for $\tilde{\mathbf{x}} \in \mathbb{R}^{N+1} \setminus \text{supp}(\tilde{p}(\tilde{\mathbf{x}}))$ and $\nabla^2 \varphi(\tilde{\mathbf{x}}) = -\tilde{p}(\tilde{\mathbf{x}})$, we conclude that $\varphi(\tilde{\mathbf{x}}) \in \mathcal{C}^2(\mathbb{R}^{N+1} \setminus \text{supp}(\tilde{p}(\tilde{\mathbf{x}})))$.

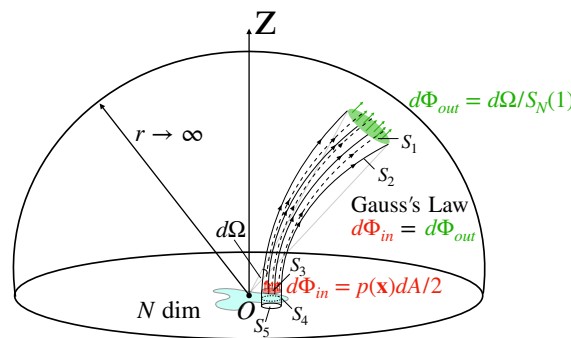

Figure 6: Proof idea of Theorem 2. By Gauss's Law, the outflow flux $d\Phi_{out}$ equals the inflow flux $d\Phi_{in}$. The factor of two in $p(\mathbf{x})dA/2$ is due to the symmetry of Poisson fields in $z < 0$ and $z > 0$.

For the uniqueness, suppose we have two different solutions $\varphi_1, \varphi_2 \in \mathcal{C}^2(\mathbb{R}^{N+1} \smallsetminus \mathrm{supp}(\tilde{p}(\tilde{\mathbf{x}})))$ which satisfy

$$\nabla^2 \varphi_1(\tilde{\mathbf{x}}) = -\tilde{p}(\tilde{\mathbf{x}}), \nabla^2 \varphi_2(\tilde{\mathbf{x}}) = -\tilde{p}(\tilde{\mathbf{x}}). \tag{12}$$

We define $\tilde{\varphi}(\tilde{\mathbf{x}}) \equiv \varphi_2(\tilde{\mathbf{x}}) - \varphi_1(\tilde{\mathbf{x}})$. Subtracting the above two equations gives

$$\nabla^2 \tilde{\varphi}(\tilde{\mathbf{x}}) = 0, \forall \tilde{\mathbf{x}} \in \mathbb{R}^{N+1} \smallsetminus \mathrm{supp}(\tilde{p}(\tilde{\mathbf{x}})). \tag{13}$$

By the vector differential identity we have

$$\tilde{\varphi}(\tilde{\mathbf{x}})\nabla^2\tilde{\varphi}(\tilde{\mathbf{x}}) = \nabla \cdot (\tilde{\varphi}(\tilde{\mathbf{x}})\nabla\tilde{\varphi}(\tilde{\mathbf{x}})) - \nabla\tilde{\varphi}(\tilde{\mathbf{x}}) \cdot \nabla\tilde{\varphi}(\tilde{\mathbf{x}}), \tag{14}$$

By the divergence theorem we have

$$\int_{\mathbb{R}^{N+1}} \nabla \cdot (\tilde{\varphi}(\tilde{\mathbf{x}})\nabla\tilde{\varphi}(\tilde{\mathbf{x}}))d^{N+1}\tilde{\mathbf{x}} = \oiint_{\partial\mathbb{R}^{N+1}} \tilde{\varphi}(\tilde{\mathbf{x}})\nabla\tilde{\varphi}(\tilde{\mathbf{x}}) \cdot d^N\mathbf{S} = 0, \tag{15}$$

where $d^N\mathbf{S}$ denotes an $N$ dimensional surface element at infinity, and the second equation holds due to zero boundary condition at infinity. Combining Eq. (13)(14)(15), we have

$$\int_{\mathbb{R}^{N+1}} \nabla \cdot (\tilde{\varphi}(\tilde{\mathbf{x}})\nabla\tilde{\varphi}(\tilde{\mathbf{x}}))d^{N+1}\tilde{\mathbf{x}} = \int_{\mathbb{R}^{N+1}\smallsetminus\mathrm{supp}(\tilde{p}(\tilde{\mathbf{x}}))} \nabla \cdot (\tilde{\varphi}(\tilde{\mathbf{x}})\nabla\tilde{\varphi}(\tilde{\mathbf{x}}))d^{N+1}\tilde{\mathbf{x}}$$

$$= \int_{\mathbb{R}^{N+1}\smallsetminus\mathrm{supp}(\tilde{p}(\tilde{\mathbf{x}}))} \|\nabla\tilde{\varphi}(\tilde{\mathbf{x}})\|^2 d^{N+1}\tilde{\mathbf{x}} = 0,$$

The first equation holds because Lebesgue measure of $\mathrm{supp}(\tilde{p}(\tilde{\mathbf{x}}))$ is zero. Since $\|\nabla\tilde{\varphi}(\tilde{\mathbf{x}})\|^2$ is an integral of a positive quantity, we must have $\nabla\tilde{\varphi}(\tilde{\mathbf{x}}) = \mathbf{0}$, or $\tilde{\varphi}(\tilde{\mathbf{x}}) = c$, $\forall\tilde{\mathbf{x}} \in \mathbb{R}^{N+1} \smallsetminus \mathrm{supp}(\tilde{p}(\tilde{\mathbf{x}}))$. This means $\varphi_1$ and $\varphi_2$ differ at most by a constant function, but a constant does not affect gradients, so $\nabla\varphi_1(\tilde{\mathbf{x}}) = \nabla\varphi_2(\tilde{\mathbf{x}})$. □

As illustrated in Fig. 6, there is a bijective mapping between the upper hemisphere of radius $r$ and the $z = 0$ plane, where each pair of corresponding points is connected by an electric field line. We will now formally prove that, in the $r \to \infty$ limit, this mapping transforms the arbitrary charge distribution in the source plane (that generated the electric field) into a uniform distribution on the hemisphere.

**Theorem 2.** *Suppose particles are sampled from a uniform distribution on the upper ($z > 0$) half of the sphere of radius $r$ and evolved by the backward ODE $\frac{d\tilde{\mathbf{x}}}{dt} = -\mathbf{E}(\tilde{\mathbf{x}})$ until they reach the $z = 0$ hyperplane, where the Poisson field $\mathbf{E}(\tilde{\mathbf{x}})$ is generated by the source $\tilde{p}(\tilde{\mathbf{x}})$. In the $r \to \infty$ limit, under the conditions in Lemma 2, this process generates a particle distribution $\tilde{p}(\tilde{\mathbf{x}})$, i.e., a distribution $p(\mathbf{x})$ in the $z = 0$ hyperplane.*

*Proof.* By Lemma 2, we know that with zero boundary at infinity, the Poisson equation $\nabla^2\varphi(\tilde{\mathbf{x}}) = -\tilde{p}(\tilde{\mathbf{x}})$ has a unique solution $\varphi(\tilde{\mathbf{x}}) \in \mathcal{C}^2$ for $\tilde{\mathbf{x}} \in \mathbb{R}^{N+1} \smallsetminus \mathrm{supp}(\tilde{p}(\tilde{\mathbf{x}}))$. Hence $\mathbf{E}(\tilde{\mathbf{x}}) = -\nabla\varphi(\tilde{\mathbf{x}}) \in \mathcal{C}^1$, guaranteeing the existence-uniqueness of the solution to the ODE $\frac{d\tilde{\mathbf{x}}}{dt} = -\mathbf{E}(\tilde{\mathbf{x}})$ according to Theorem 2.8.1 in [27].

Consider the tube in Fig. 6 connecting an area on $dA$ in the $z = \epsilon \to 0^+$ hyperplane ($S_3$) to a solid angle $d\Omega$ on the hemisphere ($S_1$), with $S_2$ as its side. The tube is the space swept by $dA$ following electric field $\mathbf{E}$, so by definition the electric field is parallel to the tangent space of the tube sides $S_2$. The bottom of the tube $S_3$ is located at $z = \epsilon \to 0^+$, a bit above the $z = 0$ plane, so the tube does not enclose any charges. We note that the divergence of Poisson field is zero in $\mathbb{R}^{N+1} \smallsetminus \mathrm{supp}(\tilde{p}(\tilde{\mathbf{x}}))$:

$$\nabla \cdot \mathbf{E}(\tilde{\mathbf{x}}) = -\nabla^2 \varphi(\tilde{\mathbf{x}}) = \tilde{p}(\tilde{\mathbf{x}}) = 0, \forall \tilde{\mathbf{x}} \in \mathbb{R}^{N+1} \smallsetminus \mathrm{supp}(\tilde{p}(\tilde{\mathbf{x}}))$$

Denote the volume and surface of the tube as $V$ and $\mathbf{B}$. According to divergence theorem, $\oiint \mathbf{E}(\tilde{\mathbf{x}}) \cdot d\mathbf{B} = \int_V \nabla \cdot \mathbf{E}(\tilde{\mathbf{x}}) dV = 0$. Hence the net flux leaving the tube is zero:

$$\Phi_{S_1} + \Phi_{S_2} + \Phi_{S_3} = 0, \quad \Phi_{S_i} \equiv \oiint_{S_i} \mathbf{E}(\tilde{\mathbf{x}}) \cdot d\mathbf{B} \quad (i = 1, 2, 3) \tag{16}$$

There is no flux through the sides, i.e., $\Phi_{S_2} = 0$, since $\mathbf{E}(\tilde{\mathbf{x}})$ is orthogonal to the surface element $d\mathbf{B}$ on the tube sides by definition. As a result, the flux $\Phi_{S_3}$ entering from below must equal the flux $\Phi_{S_1}$ leaving the other end. Denote the $l_2$ norm of the vector $\mathbf{r}$ as $r$. We first calculate the influx $\Phi_{S_3}$. To do so, we study a Gaussian pillbox whose top, side and bottom are $S_3$, $S_4$ and $S_5$. $S_3$ and $S_5$ are located at $z = \epsilon$ and $z = -\epsilon$ ($\epsilon \to 0^+$). Denote the volume and surface of the pillbox as $V'$ and $\mathbf{B}'$. The pillbox contains charge $p(\mathbf{x})dA$, so according to Gauss's law $\oiint \mathbf{E}(\tilde{\mathbf{x}}) \cdot d\mathbf{B}' = \int_{V'} \nabla \cdot \mathbf{E}(\tilde{\mathbf{x}}) dV' = \int_{V'} \tilde{p}(\tilde{\mathbf{x}}) dV' = p(\mathbf{x})dA$, i.e.,

$$\Phi'_{S_3} + \Phi'_{S_4} + \Phi'_{S_5} = p(\mathbf{x})dA, \quad \Phi'_{S_i} \equiv \oiint_{S_i} \mathbf{E}(\tilde{\mathbf{x}}) \cdot d\mathbf{B}' \quad (i = 3, 4, 5) \tag{17}$$

The flux on the sides $\Phi'_{S_4} \propto \epsilon \to 0$, and $\Phi'_{S_3} = \Phi'_{S_5}$ due to mirror symmetry of $z = 0$. So $\Phi'_{S_3} = \Phi'_{S_5} = p(\mathbf{x})dA/2$. Note on the $S_3$ surface, the outflux of the pillbox is exactly the influx of the tube, so we have:

$$\Phi_{S_3} = -\Phi'_{S_3} = -p(\mathbf{x})dA/2, \tag{18}$$

inserting which and $\Phi_{S_2} = 0$ to Eq. (16) gives

$$\Phi_{S_1} = -\Phi_{S_3} = p(\mathbf{x})dA/2. \tag{19}$$

On the other hand, in the far-field limit $r \to \infty$, since $\mathrm{supp}(p(\mathbf{x}))$ is bounded, the data distribution can be effectively seen as a point charge (see Appendix A.2). By Lemma 3, we have $\lim_{r \to \infty} \mathbf{E}(\mathbf{r}) = -\lim_{r \to \infty} \nabla \varphi(\mathbf{r}) = \frac{\mathbf{r}}{S_N(1) r^{N+1}}$. The resulting outflux on the hemisphere is

$$\Phi_{S_1} = E_r r^N d\Omega = d\Omega / S_N(1) \tag{20}$$

where $E_r \equiv \mathbf{E}(\mathbf{r}) \cdot \mathbf{r}/r$ is the radial component of $\mathbf{E}$. Comparing Eq. (19) and Eq. (20) yields $d\Omega/dA = p(\mathbf{x})S_N(1)/2 \propto p(\mathbf{x})$. In other words, the mapping from the $z = 0$ hyperplane to the hemisphere dilutes the charge density $p(\mathbf{x})$ up to a constant factor. Thus by change-of-varible, we conclude that the mapping transforms the data distribution into a uniform distribution on the infinite hemisphere. Since the ODE is reversible, the backward ODE transforms the uniform distributoin on the infinite hemisphere to the distribution $\tilde{p}(\tilde{\mathbf{x}})$. $\qquad \square$

## A.2 Multipole Expansion

We discuss the behaviors of the potential function in Poisson equation (Eq. (1)) under different scenarios, utilizing the multipole expansion. Suppose we have a unit point charge $q = 1$ located at $\mathbf{x} \in \mathbb{R}^N$. We know that the potential function at another point $\mathbf{y} \in \mathbb{R}^N$ is $\varphi(\mathbf{y} - \mathbf{x}) = 1/\|\mathbf{y} - \mathbf{x}\|^{N-2}$ (ignoring a constant factor). Now we assume that $\mathbf{x}$ is close to the origin such that we can Taylor expand around $\mathbf{x} = 0$:

$$\varphi(\mathbf{y} - \mathbf{x}) = \varphi(\mathbf{y}) - \sum_{\alpha=1}^N \mathbf{x}_\alpha \varphi_\alpha(\mathbf{y}) + \frac{1}{2} \sum_{\alpha=1}^N \sum_{\beta=1}^N \mathbf{x}_\alpha \mathbf{x}_\beta \varphi_{\alpha\beta}(\mathbf{y}) - ... \tag{21}$$

where

$$\begin{aligned}
\varphi_\alpha(\mathbf{y}) &= \left( \frac{\partial \varphi(\mathbf{y} - \mathbf{x})}{\partial \mathbf{x}_\alpha} \right)_{\mathbf{x}=0} = (N-2) \frac{\mathbf{y}_\alpha}{\|\mathbf{y}\|^N} \\
\varphi_{\alpha\beta}(\mathbf{y}) &= \left( \frac{\partial^2 \varphi(\mathbf{y} - \mathbf{x})}{\partial \mathbf{x}_\alpha \partial \mathbf{x}_\beta} \right)_{\mathbf{x}=0} = (N-2) \frac{N \mathbf{y}_\alpha \mathbf{y}_\beta - \|\mathbf{y}\|^2 \delta_{\alpha\beta}}{\|\mathbf{y}\|^{N+2}}
\end{aligned} \tag{22}$$

In the case where the source is a distribution $p(\mathbf{x})$, the potential $\varphi(\mathbf{y})$ can again be Taylor expanded:

$$\varphi(\mathbf{y}) = q\varphi(\mathbf{y}) + \sum_{\alpha=1}^{N} q_\alpha \varphi_\alpha(\mathbf{y}) + \sum_{\alpha=1}^{N} \sum_{\beta=1}^{N} q_{\alpha\beta} \varphi_{\alpha\beta}(\mathbf{y}) - ... \tag{23}$$

where

$$q = \int p(\mathbf{x})d\mathbf{x}, q_\alpha = \int p(\mathbf{x})\mathbf{x}_\alpha d\mathbf{x}, q_{\alpha\beta} = \int p(\mathbf{x})\mathbf{x}_\alpha \mathbf{x}_\beta d\mathbf{x}, \tag{24}$$

which are called monopole, dipole and quadrupole in physics, respectively. The gradient field $\mathbf{E}(y) = \nabla\Phi(\mathbf{y})$ can be expanded in the same such that

$$\mathbf{E}(\mathbf{y}) = \mathbf{E}^{(0)}(\mathbf{y}) + \mathbf{E}^{(1)}(\mathbf{y}) + \mathbf{E}^{(2)}(\mathbf{y}) + ... \tag{25}$$

It is easy to check that $\|\mathbf{E}^{(i)}(\mathbf{y})\|$ decays as $1/\|\mathbf{y}\|^{N-2+i}$, which means higher-order corrections decay faster than leading terms. So when $\|\mathbf{y}\| \to \infty$, only the monopole term $\|\mathbf{E}^{(0)}(\mathbf{y})\|$ matters, which behaves like a point source.

In a more realistic setup, we only have a large but finite $\|\mathbf{y}\|$, so the question is: under what condition is the point source approximation valid? We examine $\varphi^{(0)}$, $\varphi^{(1)}$ and $\varphi^{(2)}$ more carefully:

$$\varphi^{(0)} = \frac{1}{\|\mathbf{y}\|^{N-2}}$$
$$\varphi^{(1)} = \sum_{\alpha=1}^{N} (N-2) \frac{\mathbf{y}_\alpha \mathbf{x}_\alpha}{\|\mathbf{y}\|^N} = (N-2) \frac{\mathbf{x}^T \mathbf{y}}{\|\mathbf{y}\|^N} \tag{26}$$
$$\varphi^{(2)} = \frac{1}{2} \sum_{\alpha=1}^{N} \sum_{\beta=1}^{N} (N-2) \frac{N\mathbf{y}_\alpha \mathbf{y}_\beta - \|\mathbf{y}\|^2 \delta_{\alpha\beta}}{\|\mathbf{y}\|^{N+1}} \mathbf{x}_\alpha \mathbf{x}_\beta = \frac{N-2}{2} \frac{N(\mathbf{x}^T\mathbf{y})^2 - \|\mathbf{x}\|^2\|\mathbf{y}\|^2}{\|\mathbf{y}\|^{N+2}}$$

Since $\varphi^{(1)}$ is an odd function of $\mathbf{x}$, integrating $\varphi^{(1)}$ over $\mathbf{x}$ leads to zero (samples are normalized to zero mean). In machine learning applications, $N$ is usually a large number (although in physics $N$ is merely 3). If $\mathbf{y}$ is a random vector of length $\|\mathbf{y}\|$, then $\mathbf{x}^T\mathbf{y} \sim (\frac{1}{\sqrt{N}} \pm \frac{1}{N})\|\mathbf{x}\|\|\mathbf{y}\|$. So Eq. (26) can be approximated as

$$\varphi^{(0)} \sim \frac{1}{\|\mathbf{y}\|^{N-2}}, \varphi^{(2)} \sim \frac{\sqrt{N}}{2} \frac{\|\mathbf{x}\|^2}{\|\mathbf{y}\|^N} \tag{27}$$

Requiring $\int \varphi^{(0)}p(\mathbf{x})d\mathbf{x} \gg \int \varphi^{(2)}p(\mathbf{x})d\mathbf{x}$ gives $\|\mathbf{y}\|^2 \gg \sqrt{N}\,\mathbb{E}_{p(x)}\|\mathbf{x}\|^2$. So the condition for the point source approximation to be valid is:

$$\kappa = \frac{2\|\mathbf{y}\|^2}{\sqrt{N}\,\mathbb{E}_{p(x)}\|\mathbf{x}\|^2} \gg 1 \tag{28}$$

Based on this condition, we can partition space into three zones: (1) the far zone $\kappa \gg 1$, where the point source approximation is valid; (2) the intermediate zone $\kappa \sim O(1)$, where the gradient field has moderate curvature; (3) the near zone $\kappa \ll 1$, where the gradient field has high curvature. In practice, the initial value $\|\mathbf{y}\|$ is greater than 1000 (hence $\kappa \gg 1$) with high probability on CIFAR-10 and CelebA datasets, incidating that the initial samples lie in the far zone and gradually move toward the near zone where $\|\mathbf{y}\| \approx \|\mathbf{x}\|$ ($\kappa \ll 1$).

We summarize above observations in the following lemma in the $\|\mathbf{y}\| \to \infty$ limit:

**Lemma 3.** *Assume the data distribution $p(\mathbf{x}) \in C^0$ has a compact support in $\mathbb{R}^N$, then the solution $\varphi$ to the Poisson equation $\nabla^2\varphi(\mathbf{x}) = -p(\mathbf{x})$ with zero boundary condition at infinity satisfies $\lim_{\|\mathbf{x}\|_2 \to \infty} \nabla\varphi(\mathbf{x}) = -\frac{1}{S_{N-1}(1)} \frac{\mathbf{x}}{\|\mathbf{x}\|_2^N}$.*

*Proof.* By Lemma 1, the gradient of the solution has the following form:

$$\nabla\varphi(\mathbf{x}) = \int \nabla_{\mathbf{x}} G(\mathbf{x}, \mathbf{y})p(\mathbf{y})d\mathbf{y}, \quad \nabla_{\mathbf{x}} G(\mathbf{x}, \mathbf{y}) = -\frac{1}{S_{N-1}(1)} \frac{\mathbf{x} - \mathbf{y}}{\|\mathbf{x} - \mathbf{y}\|^N}.$$

Since $p(\mathbf{x})$ has a bounded support, we assume $\max\{\|\mathbf{x}\|_2 : p(\mathbf{x}) \neq 0\} < B$. On the other hand, we have

$$\lim_{\|\mathbf{x}\|_2 \to \infty} \nabla_{\mathbf{x}} G(\mathbf{x}, \mathbf{y}) = \lim_{\|\mathbf{x}\|_2 \to \infty} -\frac{1}{S_{N-1}(1)} \frac{\mathbf{x} - \mathbf{y}}{\|\mathbf{x} - \mathbf{y}\|^N} = \lim_{\|\mathbf{x}\|_2 \to \infty} -\frac{1}{S_{N-1}(1)} \frac{\mathbf{x}}{\|\mathbf{x}\|^N}$$

for $\forall y$ such that $\| y \|_2 < B$. Hence,

$$\lim_{\|\mathbf{x}\|_2 \to \infty} \nabla \varphi(\mathbf{x}) = \lim_{\|\mathbf{x}\|_2 \to \infty} \int \nabla_{\mathbf{x}} G(\mathbf{x}, \mathbf{y}) p(\mathbf{y}) d\mathbf{y} = \int \lim_{\|\mathbf{x}\|_2 \to \infty} \nabla_{\mathbf{x}} G(\mathbf{x}, \mathbf{y}) p(\mathbf{y}) d\mathbf{y}$$

$$= -\frac{1}{S_{N-1}(1)} \frac{\mathbf{x}}{\| \mathbf{x} \|_2^N}$$

$\square$

## A.3 Extension of Green's Function in $N$-dimensional Space

In this section, we show that the function $G(\mathbf{x}, \mathbf{y})$ defined in Eq. (2) is the $N$-dimensional extension of the Green's function, $\varphi(\mathbf{x}) = \int G(\mathbf{x}, \mathbf{y}) \rho(\mathbf{y}) d\mathbf{y}$ solves the Poisson equation $\nabla^2 \varphi(\mathbf{x}) = -\rho(\mathbf{x})$.

**Lemma 4.** *Assume the dimension $N \geq 3$, and the source term satisfies $\rho \in \mathcal{C}^0(\Omega), \int_{\mathbb{R}^N} \rho^2(\mathbf{x}) d\mathbf{x} < +\infty, \lim_{\|\mathbf{x}\|_2 \to \infty} \rho(\mathbf{x}) = 0$. The extension of Green's function $G(\mathbf{x}, \mathbf{y}) = \frac{1}{(N-2)S_{N-1}(1)} \frac{1}{\|\mathbf{x}-\mathbf{y}\|^{N-2}}$ solves the Poisson equation $\nabla_{\mathbf{x}}^2 G(\mathbf{x}, \mathbf{y}) = -\delta(\mathbf{x} - \mathbf{y})$. In addition, with zero boundary condition at infinity ($\lim_{\|\mathbf{x}\|_2 \to \infty} \varphi(\mathbf{x}) = 0$), $\varphi(\mathbf{x}) = \int G(\mathbf{x}, \mathbf{y}) \rho(\mathbf{y}) d\mathbf{y}$ solves the Poisson equation $\nabla^2 \varphi(\mathbf{x}) = -\rho(\mathbf{x})$.*

*Proof.* It is convenient to denote $\mathbf{r} = \mathbf{x} - \mathbf{y}$, $r = \|\mathbf{r}\|$ and notice $\partial r / \partial \mathbf{x} = \mathbf{r}/r$. Firstly, we calculate $\nabla_{\mathbf{x}} G(\mathbf{x}, \mathbf{y})$:

$$\nabla_{\mathbf{x}} G(\mathbf{x}, \mathbf{y}) = \frac{1}{(N-2)S_{N-1}(1)} \nabla_{\mathbf{x}} \left( \frac{1}{r^{N-2}} \right)$$

$$= \frac{1}{(N-2)S_{N-1}(1)} \frac{\partial}{\partial r} \left( \frac{1}{r^{N-2}} \right) \nabla_{\mathbf{x}} r \qquad (29)$$

$$= -\frac{1}{S_{N-1}(1)} \frac{\mathbf{r}}{r^N}$$

Then we calculate $\nabla_{\mathbf{x}}^2 G(\mathbf{x}, \mathbf{y})$:

$$\nabla_{\mathbf{x}}^2 G(\mathbf{x}, \mathbf{y}) \equiv \nabla_{\mathbf{x}} \cdot \nabla_{\mathbf{x}} G(\mathbf{x}, \mathbf{y})$$

$$= -\frac{1}{S_{N-1}(1)} \nabla_{\mathbf{x}} \cdot \frac{\mathbf{r}}{r^N}$$

$$= -\frac{1}{S_{N-1}(1)} \left( \nabla_{\mathbf{x}} \left( \frac{1}{r^N} \right) \cdot \mathbf{r} + \frac{1}{r^N} \nabla_{\mathbf{r}} \cdot \mathbf{r} \right) \qquad (30)$$

$$= -\frac{1}{S_{N-1}(1)} \left( -\frac{N}{r^N} + \frac{N}{r^N} \right)$$

$$= -\frac{0}{S_{N-1}(1) r^N}$$

which is 0 for $r > 0$, but undermined for $r = 0$. So we are left with proving

$$\int_{S_\epsilon(\mathbf{y})} \nabla_{\mathbf{x}}^2 G(\mathbf{x}, \mathbf{y}) d^N \mathbf{x} = -1, \qquad (31)$$

where $S_\epsilon(\mathbf{y})$ denotes a ball centered at $\mathbf{y}$ with a radius $\epsilon \to 0^+$. With the divergence theorem, we have

$$\int_{S_\epsilon(\mathbf{y})} \nabla_{\mathbf{x}}^2 G(\mathbf{x}, \mathbf{y}) d^N \mathbf{x} = \oiint_{\partial S_\epsilon(\mathbf{y})} \nabla_{\mathbf{x}} G(\mathbf{x}, \mathbf{y}) \cdot d^{N-1} \mathbf{B} \qquad (32)$$

where the surface integral can be computed

$$\oiint_{\partial S_\epsilon(\mathbf{y})} \nabla_{\mathbf{x}} G(\mathbf{x}, \mathbf{y}) \cdot d^{N-1} \mathbf{B} = \oiint_{\partial S_\epsilon(\mathbf{y})} \left( -\frac{1}{S_{N-1}(1)} \frac{\mathbf{r}}{r^N} \right) \cdot d^{N-1} \mathbf{B} = -\frac{1}{S_{N-1}(1)} \frac{S_{N-1}(\epsilon)}{\epsilon^{N-1}} = -1 \quad (33)$$

in which we used $\oiint_{\partial S_\epsilon(\mathbf{y})} \mathbf{r} \cdot d^{N-1} \mathbf{B} = \epsilon S_{N-1}(\epsilon)$. Together, we conclude that

$$\nabla_{\mathbf{x}}^2 G(\mathbf{x}, \mathbf{y}) = -\delta(\mathbf{x} - \mathbf{y}) \qquad (34)$$

Next we show that $\varphi(\mathbf{x}) = \int G(\mathbf{x}, \mathbf{y})\rho(\mathbf{y})d\mathbf{y}$ solves $\nabla^2 \varphi(\mathbf{x}) = -\rho(\mathbf{x})$. Taking the Laplacian operator of both sides gives:

$$
\begin{aligned}
\nabla_{\mathbf{x}}^2 \varphi(\mathbf{x}) &= \nabla_{\mathbf{x}}^2 \int G(\mathbf{x}, \mathbf{y})\rho(\mathbf{y})d\mathbf{y} \\
&= \int \nabla_{\mathbf{x}}^2 G(\mathbf{x}, \mathbf{y})\rho(\mathbf{y})d\mathbf{y} \\
&= \int -\delta(\mathbf{x} - \mathbf{y})\rho(\mathbf{y})d\mathbf{y} \quad \text{(By Eq. (34))} \\
&= -\rho(\mathbf{x})
\end{aligned}
$$

In addition, we show that $\varphi(\mathbf{x})$ is zero at infinity. Since $\rho(\mathbf{x}) \in \mathcal{C}^0$ and has compact support, we know that $\rho(\mathbf{x})$ is bounded, and let $|\rho(\mathbf{x})| < B$.

$$
\begin{aligned}
\lim_{\|\mathbf{x}\|_2 \to \infty} \varphi(\mathbf{x}) &= \lim_{\|\mathbf{x}\|_2 \to \infty} \int G(\mathbf{x}, \mathbf{y})\rho(\mathbf{y})d\mathbf{y} \\
&\leq B \lim_{\|\mathbf{x}\|_2 \to \infty} \int_{\text{supp}(\rho)} \frac{1}{(N-2)S_{N-1}(1)} \frac{1}{\|\mathbf{x} - \mathbf{y}\|^{N-2}} d\mathbf{y} \\
&= 0
\end{aligned}
$$

The last equality holds since $\text{supp}(\rho)$ is a compact set. $\qquad\square$

### A.4 Proof for the Prior Distribution on $z = z_{\text{max}}$ Hyperplane

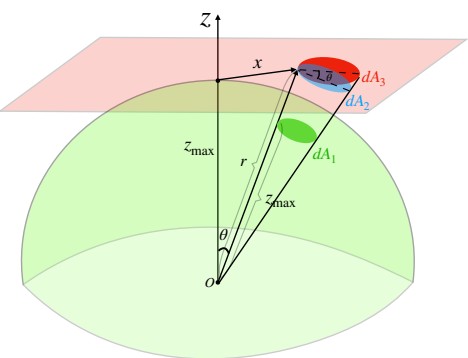

Figure 7: Diagram of the deviation in Proposition 1

We obtain the prior distribution $p_{\text{prior}}$ by projecting the uniform distribution $\mathcal{U}(S_N^+(z_{\text{max}}))$ on the hemisphere $S_N^+(z_{\text{max}})$ to the $z = z_{\text{max}}$ hyperplane. In the following proposition, we show that the projected distribution is $p_{\text{prior}}(\mathbf{x}) = \frac{2z_{\text{max}}}{S_N(1)r^{N+1}}$.

**Proposition 1.** *The radial projection of $\mathcal{U}(S_N^+(z_{max}))$ on the hemisphere $S_N^+(z_{max})$ to the $z = z_{max}$ hyperplane is $p_{prior}(\mathbf{x}) = \frac{2z_{max}}{S_N(1)r^{N+1}}$.*

*Proof.* We calculate the change-of-variable ratio by comparing two associate areas. As illustrated in Fig. 7, an area $dA_1$ on $S_N^+(z_{\text{max}})$ is projected to an area $dA_3$ on the hyperplane in the $(\mathbf{x}, z_{\text{max}})$ direction, and we have

$$
\mathcal{U}(S_N^+(z_{\text{max}}))dA_1 = p_{\text{prior}}(\mathbf{x})dA_3
$$

We aim to calculate the ratio $dA_1/dA_3$ below. We define the angle between $(\mathbf{0}, z_{\text{max}})$ and $\tilde{\mathbf{x}} = (\mathbf{x}, z_{\text{max}})$ to be $\theta$. We project $dA_3$ to the hyperplane orthogonal to $\tilde{\mathbf{x}}$ to get $dA_2 = dA_3\cos\theta = dA_3 z_{\text{max}}/r$ where $r \equiv \|\tilde{\mathbf{x}}\|_2 = \sqrt{\|\mathbf{x}\|_2^2 + z_{\text{max}}^2}$. Since $dA_1$ is parallel to $dA_2$ and they lie in the same cone from the origin $O$, we have $dA_2/dA_1 = (r/z_{\text{max}})^N$. Combining all the results gives

$$
p_{\text{prior}}(\mathbf{x}) = \mathcal{U}(S_N^+(z_{\text{max}}))\frac{dA_1}{dA_3} = \mathcal{U}(S_N^+(z_{\text{max}}))\frac{dA_1}{dA_2}\frac{dA_2}{dA_3} = \frac{2}{S_N(1)z_{\text{max}}^N}\left(\frac{z_{\text{max}}}{r}\right)^N \frac{z_{\text{max}}}{r} = \frac{2z_{\text{max}}}{S_N(1)r^{N+1}}.
$$

$\qquad\square$

In order to sample from $p_{\text{prior}}(\mathbf{x})$, we first sample the norm (radius) $R = \|\mathbf{x}\|_2$ from the distribution:

$$p_{\text{radius}}(R) \propto R^{N-1}p_{\text{prior}}(\mathbf{x}) \qquad (p_{\text{prior}} \text{ is isotropic})$$
$$\propto R^{N-1}/(\|\mathbf{x}\|_2^2 + z_{\max}^2)^{\frac{N+1}{2}}$$
$$= R^{N-1}/(R^2 + z_{\max}^2)^{\frac{N+1}{2}} \tag{35}$$

and then uniformly sample its angle. Sampling from $p_{\text{prior}}$ encompasses three steps. We first sample a real number $r_1$ with parameters $\alpha = \frac{N}{2}, \beta = \frac{1}{2}$, *i.e.*,

$$R_1 \sim \text{Beta}(\alpha, \beta)$$

Next, we set $R_2 = \frac{R_1}{1-R_1}$ such that $R_2$ is effectively sampled from the inverse beta distribution a(also known as beta prime distribution) with parameters $\alpha = \frac{N}{2}, \beta = \frac{1}{2}$. Finally, we set $R_3 = \sqrt{z_{\max}^2 R_2}$. To verify the pdf of $R_3$ is $p_{\text{radius}}$, note that the pdf of inverse beta distribution is

$$p(R_2) \propto R_2^{\frac{N}{2}-1}(1+R_2)^{-\frac{N}{2}-\frac{1}{2}}$$

Next, by change-of-variable, the pdf of $R_3 = \sqrt{z_{\max}^2 R_2}$ is

$$p(R_3) \propto R_2^{\frac{N}{2}-1}(1+R_2)^{-\frac{N}{2}-\frac{1}{2}} * \frac{2R_3}{z_{\max}^2}$$

$$\propto \frac{R_3 R_2^{\frac{N}{2}-1}}{(1+R_2)^{\frac{N+1}{2}}}$$

$$= \frac{(R_3/z_{\max})^{N-1}}{(1+(R_3^2/z_{\max}^2))^{\frac{N+1}{2}}}$$

$$\propto \frac{R_3^{N-1}}{(1+(R_3^2/z_{\max}^2))^{\frac{N+1}{2}}}$$

$$\propto \frac{R_3^{N-1}}{(z_{\max}^2 + R_3^2)^{\frac{N+1}{2}}} \propto p_{\text{radius}}(R_3) \qquad (\text{By Eq. (35)})$$

Hence we conclude that $p(R_3) = p_{\text{radius}}(R_3)$.

## B  Experimental Details

### B.1  Training

In this section we include more details about the training of PFGM and other baselines. We show the hyper-parameters settings for all the baselines (Appendix B.1.1). All the experiments are run on a single NVIDIA A100 GPU.

#### B.1.1  Additional Settings

**PFGM**  We set the hyper-parameters $\gamma = 5$, the larger batch size for calculating normalize field $|\mathcal{B}_L| = 4096$ (CIFAR-10), $256$ (CelebA), $64$ (LSUN bedroom) in Algorithm 1, and $M = 291$ (CIFAR-10, CelebA)/$356$ (LSUN bedroom), $\sigma = 0.01$ and $\tau = 0.03$ in Algorithm 2. We use the a batch size of $|\mathcal{B}| = 128$ (CIFAR-10, CelebA)/$32$ (LSUN bedroom), the same Adam optimizer and exponential moving average in [33]. We center the data around the origin. The initial $z$ components in the normalized field are approximately zero with small initial $|\epsilon_z|$ values in Algorithm 2. In this case, the trajectories of the forward ODE terminate at points that are unlikely traversed by the backward ODE, *i.e.*, points with large $\| \mathbf{x} \|_2$ and small $z$. In light of this, we heuristically confine the maximum sampling step to $M = 200$ (CIFAR-10, CelebA)/$250$ (LSUN bedroom) for points with the initial $|\epsilon_z|$ smaller than $0.005$. More principal solutions are left for future works.

For *selecting $M$ in more general settings*, we recommend the following rule-of-thumb. According to analysis in Section A.2, given a perturbation point $(\mathbf{y}, z)$ when setting the exponent $m = M$ in Algorithm 2, we can ensure the point source approximation by

$$\|\mathbf{y}\|^2 \gg \sqrt{N}\mathbb{E}_{p(\mathbf{x})}\|\mathbf{x}\|^2/2 \tag{36}$$

where $N$ is the data dimension and $p(\mathbf{x})$ is the data distribution. By WLLN, we have $\|\epsilon_{\mathbf{x}}\| = \sqrt{N}\sigma$, and recall that $\mathbf{y} = \mathbf{x} + \|\epsilon_{\mathbf{x}}\|(1+\tau)^M \mathbf{u}$ where $\epsilon = (\epsilon_{\mathbf{x}}, \epsilon_z) \sim \mathcal{N}(0, \sigma^2 I_{N+1 \times N+1})$, $\mathbf{u} \sim \mathcal{U}(S_N(1))$. Together, we conclude $\|\mathbf{y}\| \approx \sqrt{N}\sigma(1+\tau)^M$. Substituting in Eq. (36), we have

$$M > \frac{1}{2}\log_{1+\tau}\frac{\mathbb{E}_{p(\mathbf{x})}\|\mathbf{x}\|^2}{2\sqrt{N}\sigma^2} = \frac{1}{2}\frac{\ln\frac{\mathbb{E}_{p(\mathbf{x})}\|\mathbf{x}\|^2}{2\sqrt{N}\sigma^2}}{\ln 1+\tau}$$

We empirically observe that setting $M = \frac{3}{4}\frac{\ln\frac{\mathbb{E}_{p(\mathbf{x})}\|\mathbf{x}\|^2}{2\sqrt{N}\sigma^2}}{\ln 1+\tau}$ already gives good results, and the corresponding $\|\mathbf{y}\| \approx 3000$. For example, on CIFAR-10 datasets, $N = 3072, \tau = 0.03, \sigma = 0.01, \mathbb{E}_{p(\mathbf{x})}\|\mathbf{x}\|^2 \approx 900$, we have $M = \frac{3}{4}\frac{\ln\frac{\mathbb{E}_{p(\mathbf{x})}\|\mathbf{x}\|^2}{2\sqrt{N}\sigma^2}}{\ln 1+\tau} \approx 291$.

Since we are operating in the augmented space, we add minor modifications to the DDPM++/DDPM++ deep architectures to accommodate the extra dimension. More specifically, we replace the conditioning time variable in VP/sub-VP with the additional dimension $z$ in PFGM as the input to the positional embedding. We also need to add an extra scalar output representing the $z$ direction. To this end, we add an additional output channel to the final convolution layer and take the global average pooling of this channel to obtain the scalar. For LSUN bedroom dataset, we both experiments with the channel configurations suggested in NSCN++ [33] and DDPM [16].

**VE/VP/sub-VP** We use the same set of hyper-parameters and the NCSN++/DDPM++ (deep) backbone and the continuous-time training objectives for forward SDEs in [33].

## B.2 Sampling

We provide more details of PFGM and VE/VP sampling implementations in Appendix B.2.1. We further discuss two techniques used in PFGM ODE sampler: change-of-variable formula (Appendix B.2.2) and the substitution of ground-truth Poisson field direction on $z$ (Appendix B.2.3).

### B.2.1 Additional settings

**PFGM** For RK-45 sampler, we use the function implemented in `scipy.integrate.solve_ivp` with `atol`$=1e-4$, `rtol`$=1e-4$. For forward Euler method, we discretize the ODE with constant step size determined by the number of steps, *i.e.*, step size $= (\log z_{\max} - \log z_{\min})$/number of steps for the backward ODE (Eq. (6)). As in [1], we set the terminal value of $z_{min} = 1e-3$. We choose $z_{\max} = 40$ (CIFAR-10), $60$ (CelebA $64^2$), $100$ (LSUN bedroom) to satisfy the condition $\kappa \gg 1$ by the multipole expansion analysis in Appendix A.2. The condition ensures that the data distribution can be viewed roughly as a point source at origin. For example, we set $z_{max} = 40$ on CIFAR-10, and the corresponding $\kappa$ is greater than $50$ with high probability. The hyperparameters work well without further fine tuning. Hence, we hypothesize that PFGM is insensitive to the choice of hyperparameters in a reasonable range, as shown in Table 3. We clip the norms of initial samples into $(0, 3000)$ for CIFAR-10, $(0, 6000)$ for CelebA and $(0, 30000)$ for LSUN bedroom.

For selecting $z_{\max}$ and clipping upper bound of norms for general datasets, we recommend the following rule-of-thumb. Recall that during the training perturbations (Eq. (5)), given a random initial value $\epsilon_z \sim \mathcal{N}(0, \sigma^2)$, maximum $z$ is

$$z = |\epsilon_z|(1+\tau)^M$$

Hence we set $z_{\max} = \mathbb{E}[|\epsilon_z|(1+\tau)^M] = \sqrt{\frac{2}{\pi}}\sigma(1+\tau)^M$. For example, on CIFAR-10, $\tau = 0.03, M = 291$, and $z_{\max} \approx 43$. The clipping upper value is similarity derived, by setting it to $\mathbb{E}[\|\epsilon_{\mathbf{x}}\|(1+\tau)^M] = \sqrt{N}\sigma(1+\tau)^M \approx 3000$, where $\epsilon_{\mathbf{x}} \sim \mathcal{N}(0, \sigma^2 I_{N \times N})$. By combining Eq. (36), we further have

$$z_{\max} = \sqrt{\frac{2}{\pi}}\sigma(1+\tau)^M = \sqrt{\frac{2}{\sigma\pi}}\left(\frac{\mathbb{E}_{p(\mathbf{x})}\|\mathbf{x}\|^2}{2\sqrt{N}}\right)^{\frac{3}{4}}$$

$$\text{clipping upper value} = \sqrt{N}\sigma(1+\tau)^M = \sqrt{\frac{N}{\sigma}}\left(\frac{\mathbb{E}_{p(\mathbf{x})}\|\mathbf{x}\|^2}{2\sqrt{N}}\right)^{\frac{3}{4}}$$

where $N$ is the data dimension and $p(\mathbf{x})$ is the data distribution. These formulas are easier for practitioner to apply PFGM on new datasets.

**VE/VP/sub-VP** For the PC sampler in VE, we follow [33] to set the reverse diffusion process as the predictor and the Langevin dynamics (MCMC) as the corrector. For VP/sub-VP, we drop the corrector in PC sampler since it only gives slightly better results [33].

Table 3: FID scores versus $z_{max}$ on PFGM w/ DDPM++

| $z_{max}$ | 30 | 40 | 50 |
|---|---|---|---|
| **FID score** | 2.49 | 2.48 | 2.48 |

### B.2.2 Exponential Decay on $z$ Dimension

Recall that in Section 3.3, we replace the vanilla backward ODE with a new ODE anchored by $z$:

$$d(\mathbf{x}, z) = (\frac{d\mathbf{x}}{dt}\frac{dt}{dz}dz, dz) = (\mathbf{v}(\tilde{\mathbf{x}})_{\mathbf{x}}\mathbf{v}(\tilde{\mathbf{x}})_z^{-1}, 1)dz$$

We further use the change-of-variable formula, *i.e.*, $t' = -\log z$, to achieve exponential decay on the $z$ dimension:

$$d(\mathbf{x}, z) = (\mathbf{v}(\tilde{\mathbf{x}})_{\mathbf{x}}\mathbf{v}(\tilde{\mathbf{x}})_z^{-1}z, z)dt'$$

The trajectories of the two ODEs above are the same when $dt, dt' \to 0$. We compare the NFE and the sample quality of different ODEs in Table 4. We measure the NFE/FID of generating 50000 CIFAR-10 samples with the RK45 method in Scipy package [37]. The batch size is set to 1000. All the numbers are produced on a single NVIDIA A100 GPU. We observe that the ODE with the anchor variable $t'$ not only accelerates the vanilla by 2 times, but has almost no harm to the sample quality measured by FID score.

Table 4: NFE and FID scores of different backward ODEs in PFGM

| **Algorithm** | $d(\mathbf{x}, z)/dz$ | $d(\mathbf{x}, z)/dt'$ |
|---|---|---|
| **NFE** | 242 | 104 |
| **FID score** | 2.53 | 2.48 |

### B.2.3 Substitute the Predicted $z$ Direction with the Ground-truth

Since the neural network cannot perfectly learn the ground-truth $z$ direction, we replace the predicted $f_\theta(x)_z$ with the ground-truth direction when $z$ is small. More specifically, given $\tilde{\mathbf{x}} = (\mathbf{x}, z) \in \mathbb{R}^{N+1}$, recall that the empirical field is $\hat{\mathbf{E}}(\tilde{\mathbf{x}}) = c(\tilde{\mathbf{x}})\sum_{i=1}^{n}\frac{\tilde{\mathbf{x}}-\tilde{\mathbf{x}}_i}{\|\tilde{\mathbf{x}}-\tilde{\mathbf{x}}_i\|^{N+1}}$ where $c(\tilde{\mathbf{x}}) = 1/\sum_{i=1}^{n}\frac{1}{\|\tilde{\mathbf{x}}-\tilde{\mathbf{x}}_i\|^{N+1}}$. Hence we can rewrite the empirical field as

$$\hat{\mathbf{E}}(\tilde{\mathbf{x}}) = \sum_{i=1}^{n} w(\tilde{\mathbf{x}}, \tilde{\mathbf{x}}_i)(\tilde{\mathbf{x}} - \tilde{\mathbf{x}}_i)$$

where $\sum_{i=1}^{n} w(\tilde{\mathbf{x}}, \tilde{\mathbf{x}}_i) = \sum_{i=1}^{n} \frac{\frac{1}{\|\tilde{\mathbf{x}}-\tilde{\mathbf{x}}_i\|^{N+1}}}{\sum_{j=1}^{n}\frac{1}{\|\tilde{\mathbf{x}}-\tilde{\mathbf{x}}_j\|^{N+1}}} = 1$. Furthermore we have $\forall i, (\tilde{\mathbf{x}} - \tilde{\mathbf{x}}_i)_z = z - 0 = z$.

Together, the $z$ component in the empirical field is $\hat{\mathbf{E}}(\tilde{\mathbf{x}})_z = \sum_{i=1}^{n} w(\tilde{\mathbf{x}}, \tilde{\mathbf{x}}_i)(\tilde{\mathbf{x}} - \tilde{\mathbf{x}}_i)_z = z$. The predicted normalized field (on $\mathbf{x}$) is trained to approximate the normalized field (on $\mathbf{x}$), *i.e.*,

$$f_\theta(\tilde{\mathbf{x}})_{\mathbf{x}} \approx -\sqrt{N}\hat{\mathbf{E}}(\tilde{\mathbf{x}})_{\mathbf{x}}/(\sqrt{\|\hat{\mathbf{E}}(\tilde{\mathbf{x}})_{\mathbf{x}}\|_2^2 + z^2} + \gamma)$$

$$\approx -\sqrt{N}\hat{\mathbf{E}}(\tilde{\mathbf{x}})_{\mathbf{x}}/(\sqrt{\|\hat{\mathbf{E}}(\tilde{\mathbf{x}})_{\mathbf{x}}\|_2^2} + \gamma)$$

The last approximation is due to $\|\hat{\mathbf{E}}(\tilde{\mathbf{x}})_{\mathbf{x}}\|_2 \gg z$. Solving for $\|\hat{\mathbf{E}}(\tilde{\mathbf{x}})_{\mathbf{x}}\|_2$, we get $\|\hat{\mathbf{E}}(\tilde{\mathbf{x}})_{\mathbf{x}}\|_2 \approx \frac{\gamma\|f_\theta(\tilde{\mathbf{x}})_{\mathbf{x}}\|_2/\sqrt{N}}{1-\|f_\theta(\tilde{\mathbf{x}})_{\mathbf{x}}\|_2/\sqrt{N}}$. Hence the $z$ component in the normalized field after substituting the ground-truth

is $\hat{\mathbf{E}}(\tilde{\mathbf{x}})_z/(\sqrt{\|\hat{\mathbf{E}}(\tilde{\mathbf{x}})_{\mathbf{x}}\|_2^2 + z^2} + \gamma) = z/(\sqrt{(\frac{\gamma\|f_\theta(\tilde{\mathbf{x}})_{\mathbf{x}}\|_2/\sqrt{N}}{1-\|f_\theta(\tilde{\mathbf{x}})_{\mathbf{x}}\|_2/\sqrt{N}})^2 + z^2} + \gamma)$. In our experiments, we therefore replace the original prediction $f_\theta(\tilde{\mathbf{x}})_z$ with $-\sqrt{N}z/(\sqrt{(\frac{\gamma\|f_\theta(\tilde{\mathbf{x}})_{\mathbf{x}}\|_2/\sqrt{N}}{1-\|f_\theta(\tilde{\mathbf{x}})_{\mathbf{x}}\|_2/\sqrt{N}})^2 + z^2} + \gamma)$ when $z < 5/1/0.1$ during the backward ODE sampling for CIFAR-10/CelebA $64^2$/LSUN bedroom $256^2$.

Table 5 reports the NFE and FID score w/o and w/ the above substitution. We observe that the usage of ground-truth $z$ direction in the near field accelerates the sampling speed.

Table 5: NFE and FID scores of w/ and w/o substitution

| Algorithm | w/o substitution | w/ substitution |
|---|---|---|
| **NFE** | 134 | 104 |
| **FID score** | 2.48 | 2.48 |

### B.3 Evaluation

We use FID [13] and Inception scores [29] to quantitatively measure the sample quality, and NFE (number of evaluation steps) for the inference speed. FID (Fréchet Inception Distance) score is the Fréchet distance between two multivariate Gaussians, whose means and covariances are estimated from the 2048-dimensional activations of the Inception-v3 [34] network for real and generated samples respectively. Inception score is the exponential mutual information between the predicted labels of the Inception network and the images. We also report bits/dim for likelihood evaluation. It is computed by dividing the negative log-likelihood by the data dimension, *i.e.*, bits/dim = $-\log p_{\text{prior}}(\mathbf{x})/N$.

For CIFAR-10, we compute the Fréchet distance between 50000 samples and the pre-computed statistics of CIFAR-10 dataset in [13]. For CelebA $64 \times 64$, we follow the setting in [32] where the distance is computed between 10000 samples and the test set. For model selection, we follow [32] and pick the checkpoint with smallest FID every 50k iterations on 10k samples for computing all the scores.

### B.4 Effects of Step Size: FID versus NFE

For preciseness, Table 6 reports the exact numbers in Fig. 5(c).

Table 6: The FID scores in Fig. 5(c) of different methods and NFE.

| Method / NFE | 10 | 20 | 50 | 100 |
|---|---|---|---|---|
| **VP-ODE** | 192.36 | 72.25 | 38.18 | 19.73 |
| **DDIM** | 13.36 | 6.48 | 4.67 | 4.16 |
| **PFGM** | 14.98 | 6.46 | 3.48 | 2.89 |

Since in the ODE $d(\mathbf{x}, z) = -(\mathbf{v}(\tilde{\mathbf{x}})_{\mathbf{x}}\mathbf{v}(\tilde{\mathbf{x}})_z^{-1}z, z)dt'$ of PFGM, the $z$ variable is a function of $t'$ ($z = e^{t'}$), we integrate the $z$ in the Euler method to reduce the discretization error. The vanilla update from time $t'_i$ to time $t'_{i+1}$ is $(\mathbf{x}_{i+1}, z_{i+1}) = (\mathbf{x}_i, z_i) - (\mathbf{v}(\tilde{\mathbf{x}}_i)_{\mathbf{x}}\mathbf{v}(\tilde{\mathbf{x}}_i)_{z_i}^{-1}z_i, z_i)(t'_{i+1} - t'_i)$, and the new update is $(\mathbf{x}_{i+1}, z_{i+1}) = (\mathbf{x}_i, z_i) - (\mathbf{v}(\tilde{\mathbf{x}}_i)_{\mathbf{x}}\mathbf{v}(\tilde{\mathbf{x}}_i)_{z_i}^{-1}\int_{t'_i}^{t'_{i+1}} z(t')dt', \int_{t'_i}^{t'_{i+1}} z(t')dt')$. We empirically observe that the new update scheme significantly improve the FID score.

## C Failure of VE/VP-ODE on NCSNv2 backbone

In Fig. 5(a), we demonstrate the trajectories of cleaner samples/noisier samples/noisier samples w/ corrector. We visualize these three groups in Fig. 8(a) and Fig. 8(b). The noisier samples are marked with red boxes in Fig. 8(a) and the remaining images in Fig. 8(a) are cleaner samples. The samples within green boxes in Fig. 8(b) are noisier samples w/ corrector. Samples on the same spatial locations in the two figures are generated by identical initial latents.

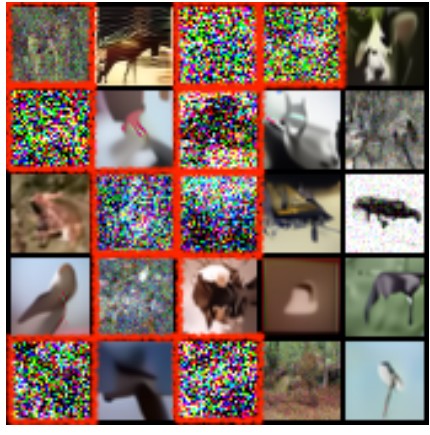 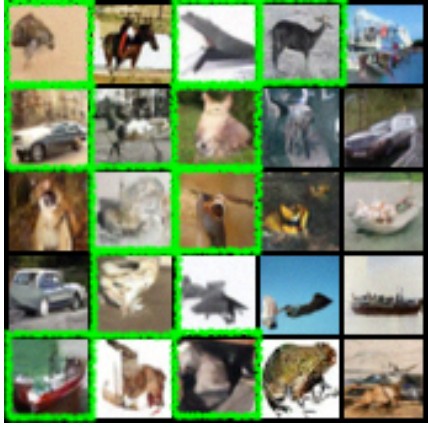

(a) Samples from VE-ODE (Euler)          (b) Samples from VE-ODE (Euler w/ corrector)

Figure 8: **(a)** Samples from VE-ODE (Euler w/o corrector). We highlight the noisier images with red boxes. The rest are cleaner images. **(b)** Samples from VE-ODE (Euler w/ corrector). We mark the noisier samples after correction with green boxes.

The Gaussian kernels in score-based models are $\mathcal{N}(\mathbf{x}, \sigma(t)^2)$ (VE) and $\mathcal{N}(\sqrt{1-\sigma(t)^2}\mathbf{x}, \sigma(t)^2)$ (VP) [33]. When $\sigma(t)$ is large, the norms of perturbed samples are approximately $\sqrt{N}\sigma(t)$. The backward ODE could break down if the trajectories diverge from the norm-$\sigma(t)$ relation, as shown by the noisier samples' trajectories in Fig. 5(a). In contrast, the norm distributions of PFGM is approximately $p(\parallel \mathbf{x} \parallel) \propto \parallel \mathbf{x} \parallel_2^{N-2}/(\parallel \mathbf{x} \parallel_2^2 + z^2)^{\frac{N}{2}}$ when $z$ is large (see deviation for $p_{\text{prior}}$ in Appendix A.4), which have a wider span for high density region (see Fig. 4). The weak correlation between norm and $z$ makes PFGM more robust on the lighter NCSNv2 backbone.

# D   Extra Experiments

## D.1   LSUN Bedroom $256 \times 256$

We report the FID scores and NFEs for LSUN bedroom dataset in Table 7. We adopt the code base of [33] in our experiments. In [33], they experimented on the LSUN bedroom $256 \times 256$ dataset only on VE-SDE using a deeper NCSN++ backbone. In our DDPM++ architecture, we directly borrow the configuration of channels from the NCSN++ architecture [33] in each residual block (PFGM w/ NCSN++ channel). We further change $z_{max}$ to 100, as it empirically gives better sample quality.

We also evaluate the performance when using the configuration of channels in the DDPM [16] architecture (PFGM w/ DDPM channel). We use the RK45 [7] solver in the Scipy library [37] for PFGM sampling. We report the FID score using the evaluation protocol in [5].

Table 7: FID/NFE on LSUN bedroom $256 \times 256$

|                        | FID ↓  | NFE ↓ |
|------------------------|--------|-------|
| StyleGAN [18]          | **2.65** | **1** |
| DDPM [16]              | 6.86   | 1000  |
| VE-SDE [33]            | 11.75  | 2000  |
| PFGM w/ NCSN++ channel | 17.01  | 134   |
| PFGM w/ DDPM channel   | 13.66  | 122   |

Table 7 shows that PFGM has comparable performance with VE-SDE when using DDPM channel, while achieving around 15× acceleration. We observe that PFGM achieves a better FID score

using the similar configuration in the DDPM model, and converges faster — 150k over the total 2.4M training iterations suggested in [33]. Remarkably, the VE-ODE baseline — the method most comparable to ours — only produces noisy samples on this dataset. It suggests that PFGM is able to scale up to high resolution images when using advanced architectures. We also compare with the number reported in [16] using similar architecture. Note that DDPM requires 1000 NFE during sampling, and doesn't possess invertibility compared to flow models.

## D.2 Results on NCSNv2 Architecture

In this section, we demonstrate the image generation on CIFAR-10 and CelebA $64 \times 64$, using NCSNv2 architecture [32], which is the predecessor of NCSN++ and DDPM++ [33] and has smaller capacity. Since the VE/VP-ODE has poor performance (FID greater than 90), with the RK45 solver, we also apply the forward Euler method (**Euler**) with fixed number of steps. We explicitly name the sampler, with forward Euler method as predictor and Langevin dynamics as corrector, as **Euler w/ corrector**. For Euler w/ corrector in VE/VP-ODE, we use the probability flow ODE (reverse-time ODE) as the predictor and the Langevin dynamics (MCMC) as the corrector. We borrow all the hyper-parameters from [33] except for the signal-to-noise ratio. We empirically observe the new configurations in Table 8 give better results on the NCSNv2 architecture.

To accommodate the extra dimension $z$ on NCSNv2, we concatenate the image with an additional constant channel with value $z$ and thus the first convolution layer takes in four input channels. We also add an additional output channel to the final convolution layer and take the global average pooling of this channel to obtain the direction on $z$.

Table 8: Signal-to-noise ratio of different dataset-method pairs

| Dataset-Method | CIFAR-10 - VE | CIFAR-10 - VP | CelebA - VE | CelebA - VP |
|---|---|---|---|---|
| **signal-to-noise ratio** | 0.16 | 0.27 | 0.12 | 0.27 |

### D.2.1 CIFAR-10

Table 9 reports the image quality measured by Inception/FID scores and the inference speed measured by NFE on CIFAR-10, using a weaker architecture NCSNv2 [32]. We show that PFGM with the RK45 solver has competitive FID/Inception scores with the Langevin dynamics, which was the best model on the NCSNv2 architecture before, and requires $10\times$ less NFE. In addition, PFGM performs better than all the other ODE samplers. Our method is more tolerant of sampling error. Among the compared ODEs, our backward ODE (Eq. (6)) is the only one that successfully generates high quality samples while the VE/VP-ODE fail w/o the Langevin dynamics corrector. The backward ODE still beats the baselines w/ corrector.

### D.2.2 CelebA

In Table 10, we report the quality of images generated by models trained on CelebA $64 \times 64$, as measured by the FID scores, and the sampling speed, as measured by NFE. We use this dataset as our preliminary experiments hence we only apply NCSNv2 [32] for different baselines. As shown in Table 10, PFGM achieves best FID scores than all the baselines on CelebA dataset, while accelerating the inference speed around $20\times$. Remarkably, PFGM outperforms the Langevin dynamics and reverse-time SDE samplers, which are usually considered better than their deterministic counterparts.

**Remark: On the FID scores on CelebA** $64 \times 64$   One interesting observation is that the samples of PFGM (RK45) (Fig. 9(b)) contain more obvious artifacts than Langevin dynamics (Fig. 9(a)), although PFGM has a lower FID score on the same architecture. We hypothesize that the diversity of samples has larger effects on the FID scores than the artifacts. As shown in Fig. 9(a) and Fig. 9(b), samples generated by PFGM have more diverse background colors and hair colors than samples of Langevin dynamics. In addition, we evaluate the performance of PFGM on the DDPM++ architecture. We show that the FID score can be further reduced to $3.68$ using the more advanced DDPM++ architecture. By examining the generated samples of PFGM on DDPM++ (Fig. 13), we observe that the samples are diverse and exhibit fewer artifacts than PFGM on NCSNv2. It suggests that by using

Table 9: CIFAR-10 sample quality (FID, Inception) and number of function evaluation (NFE). All the methods below the *NCSNv2 backbone* separator use the NCSNv2 [32] network architecture as the backbone.

|  | Inception ↑ | FID ↓ | NFE ↓ |
|---|---|---|---|
| PixelCNN [36] | 4.60 | 65.93 | 1024 |
| IGEBM [8] | 6.02 | 40.58 | 60 |
| WGAN-GP [12] | 7.86 ± .07 | 36.4 | 1 |
| SNGAN [26] | 8.22 ± .05 | 21.7 | 1 |
| NCSN [31] | **8.87 ± .12** | 25.32 | 1001 |
| *NCSNv2 backbone* | | | |
| Langevin dynamics [32] | 8.40 ± .07 | **10.87** | 1161 |
| VE-SDE [33] | 8.23 ± .02 | 10.94 | 1000 |
| VP-SDE [33] | 6.85 ± .01 | 44.05 | 1000 |
| VE-ODE (Euler w/ corrector) | 8.05 ± .03 | 11.33 | 1000 |
| VP-ODE (Euler w/ corrector) | 7.33 ± .07 | 37.74 | 1000 |
| PFGM (Euler) | 8.00 ± .09 | 11.78 | 200 |
| PFGM (RK45) | 8.30 ± .05 | 11.22 | **118** |

a more powerful architecture like DDPM++, we can remove the artifacts while retaining the diversity in PFGM.

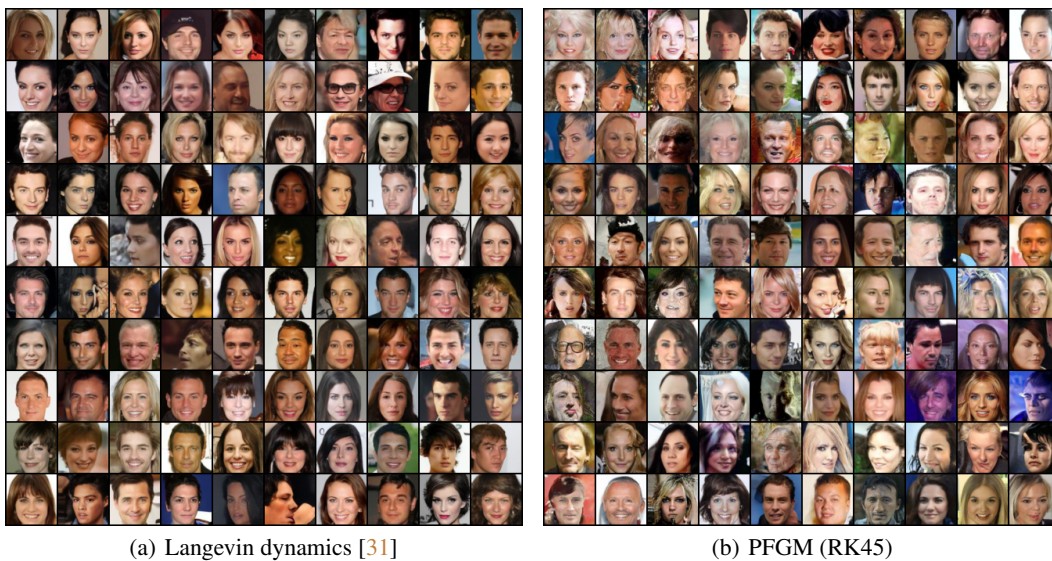

(a) Langevin dynamics [31]    (b) PFGM (RK45)

Figure 9: Uncurated samples from Langevin dynamics [31] and PFGM (RK45), both using the NCSNv2 architecture.

### D.3 Wall-clock Sampling Time

The main bottleneck of sampling time in each ODE step is the function evaluation of the neural network. Hence, for different ODE equations using similar neural network architectures, their inference times per ODE step are approximately the same.

We implement PFGM on the NCSNv2 [32], DDPM++ [33], and DDPM++ deep [33] architectures, with sight modifications to account for the extra dimension $z$. In Table 11, we report the sampling time per ODE step method with the DDPM++ backbone, as well as the total sampling time. We measure the sampling time of generating a batch of 1000 images on CIFAR-10. We compare PFGM, VP/sub-VP ODEs using the RK45 solver. As a reference, we also report the results of VP-SDE using the predictor-corrector sampler [33]. All the numbers are produced on a single NVIDIA A100 GPU.

Table 10: FID/NFE on CelebA $64 \times 64$

|  | FID ↓ | NFE ↓ |
|---|---|---|
| NCSN [31] | 26.89 | 1001 |
| *NCSNv2 backbone* | | |
| Langevin dynamics [32] | 10.23 | 2501 |
| VE-SDE [33] | 8.15 | 1000 |
| VP-SDE [33] | 34.52 | 1000 |
| VE-ODE (Euler w/ corrector) | 8.30 | 200 |
| VP-ODE (Euler w/ corrector) | 41.81 | 200 |
| PFGM (Euler) | 7.85 | **100** |
| PFGM (RK45) | 7.93 | 110 |
| *DDPM++ backbone* | | |
| PFGM (RK45) | **3.68** | 110 |

Table 11: Wall-clock sampling time (second)

| **Method** | PFGM | VP-ODE | sub-VP-ODE | VP-SDE (PC) |
|---|---|---|---|---|
| **NFE** | 110 | 134 | 146 | 1000 |
| **Wall-clock time per step** | 0.526 | 0.522 | 0.520 | 0.491 |
| **Total wall-clock time** | 57.81 | 69.97 | 75.92 | 490.65 |

As expected, ODEs using similar architectures and the same solver have nearly the same wall-clock time per ODE step. The table also shows that PFGM achieves the smallest total wall-clock sampling time.

### D.4 Image Interpolations

The invertibility of the ODE in PFGM enables the interpolations between pairs of images. As shown in Fig. 10, we adopt the spherical interpolations between the latent representations of the images in the first and last column.

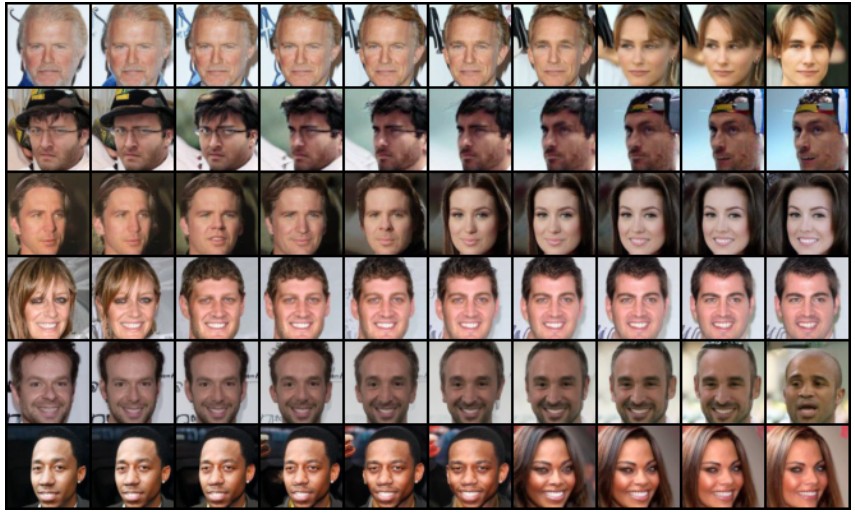

Figure 10: Interpolation on CelebA $64 \times 64$ by PFGM

### D.5 Temperature Scaling

To demonstrate more utilities of the meaningful latent space of PFGM, we include the experiments of temperature scaling on CelebA $64 \times 64$ dataset. We linearly increase the norm of latent codes from 1000 to 6000 to get the samples in Fig. 11.

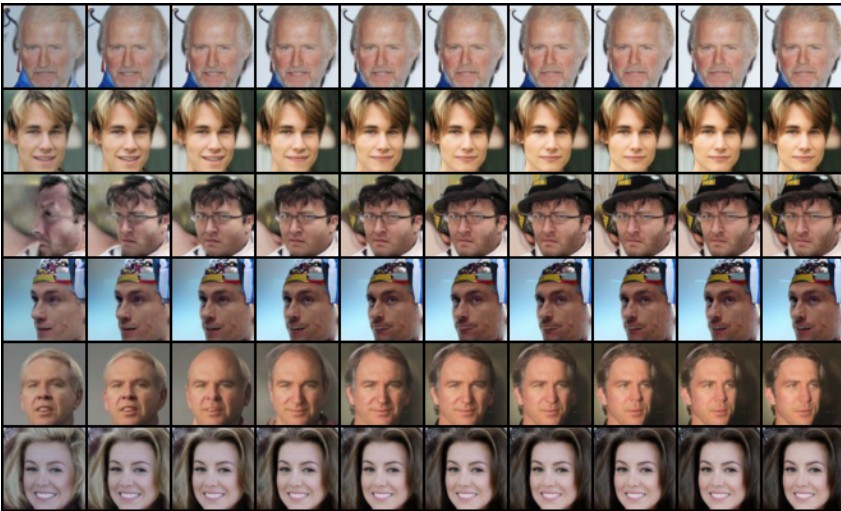

Figure 11: Temperature scaling on CelebA $64 \times 64$ by PFGM

## E  Extended Examples

We provide extended samples from PFGM on CIFAR-10 (Fig. 12), CelebA $64 \times 64$ (Fig. 13) and LSUN bedroom $256 \times 256$ (Fig. 14) datasets.

## F  Physical Interpretation of the ODEs in PFGM

In Section 2, in order to move the particles along the electric lines, we set the time derivative of $x$ to the Poisson field $\mathbf{E}(x)$:

$$[q = 1, \text{forward ODE}] \quad \frac{d\mathbf{x}}{dt} = \mathbf{E}(\mathbf{x}), \quad [q = -1, \text{backward ODE}] \quad \frac{d\mathbf{x}}{dt} = -\mathbf{E}(\mathbf{x}) \qquad (37)$$

We give the interpretation of the ODEs from a physical perspective. Newton's law implies that the external force is proportional to the acceleration of the particle. In the overdamped limit, e.g., when the particle is moving in honey, the external force is instead proportional to the velocity of the particle, making the equation of motion a first-order ODE. Denoting the viscosity of the fluid as $\gamma$, the dynamics of the particle under the influence of the electric field of the source $\rho(\mathbf{x})$ is

$$m\frac{d^2\mathbf{x}}{dt^2} = -\gamma\frac{d\mathbf{x}}{dt} + q\mathbf{E}(\mathbf{x}),$$

which has an overdamped limit $\frac{d\mathbf{x}}{dt} = q\mathbf{E}(\mathbf{x})$ when we set $t \to \gamma t$ and $\gamma \to \infty$. In this case, a particle with mass $m = 1$ and charge $q = 1$ would follow the electric field with velocity equal to $\mathbf{E}$, justifying Eq. (37).

## G  Limitations and Future Directions

In Section 3.2 we discuss the training paradigm of PFGM, including the normalized Poisson field and the discretized forward ODE. There are several potential improvements. First, the normalized field on mini-batch is biased. In this paper, we directly alleviate the bias by using a larger training batch. However, it does not solve the problem fundamentally. Some potential directions are incorporating

more physical tools: we can exploit renormalization to make the Poisson field well-behaved in near fields. Another possibility is to replace a point charge with a quantum particle, whose position uncertainty fills the empty space among nearest neighbor data samples and makes the data manifold smoother.

## H  Potential Social Impact

Generative models is a rapidly growing field of study with far-reaching implications for science and society. Our work proposes a new generative model that allows for high-quality samples, quick inference and adaptivity. Many downstream applications benefit from our PFGM models' powerful expressive capabilities, particularly those that need fast inference speed and good sample quality at the same time. The usage of these models might have both positive and negative outcomes depending on the downstream use case. For example, PFGM can be incorporated in producing good image/audio samples by the fast backward ODE. This, on the other hand, promotes *deepfake* technology and leads to social scams. Generative models are also brittle and susceptible to backdoor adversarial attacks on publicly available training data, causing unanticipated failure. Addressing the above concerns requires further research in providing robustness guarantees for generative models as well as close collaborations with researchers in socio-technical disciplines.

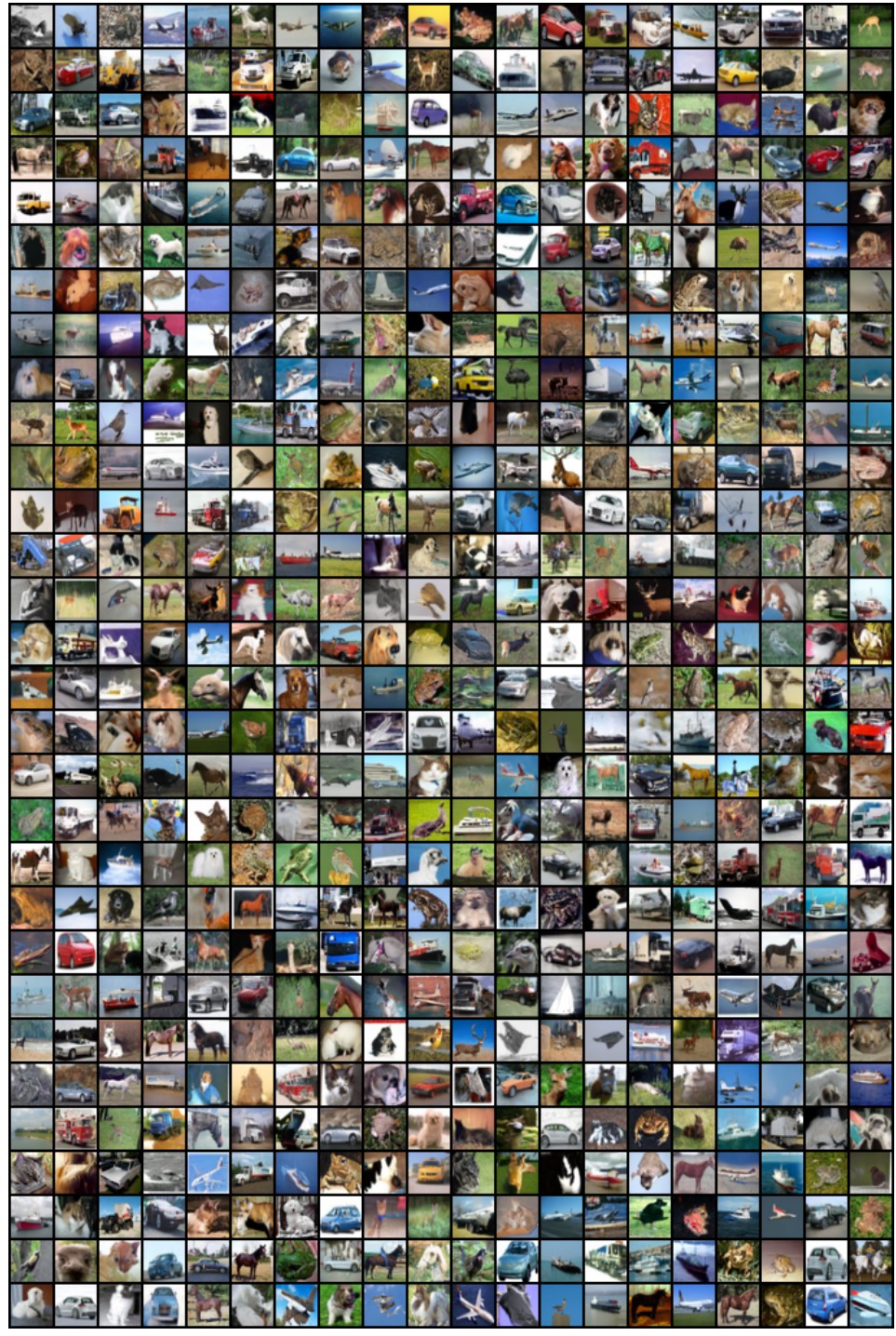

Figure 12: CIFAR-10 samples from PFGM (RK45)

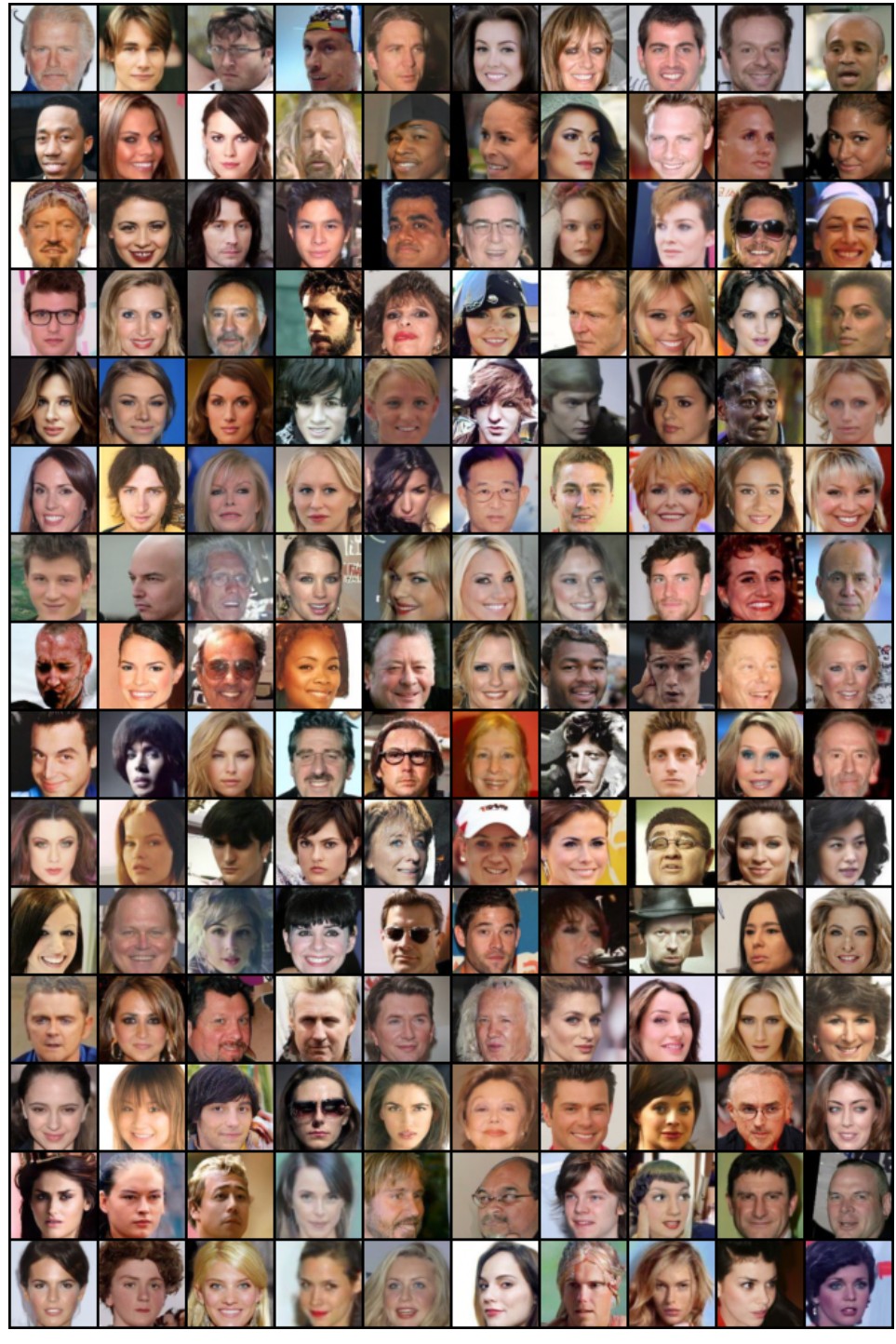

Figure 13: CelebA $64 \times 64$ samples from PFGM (RK45, NCSNv2 architecture)

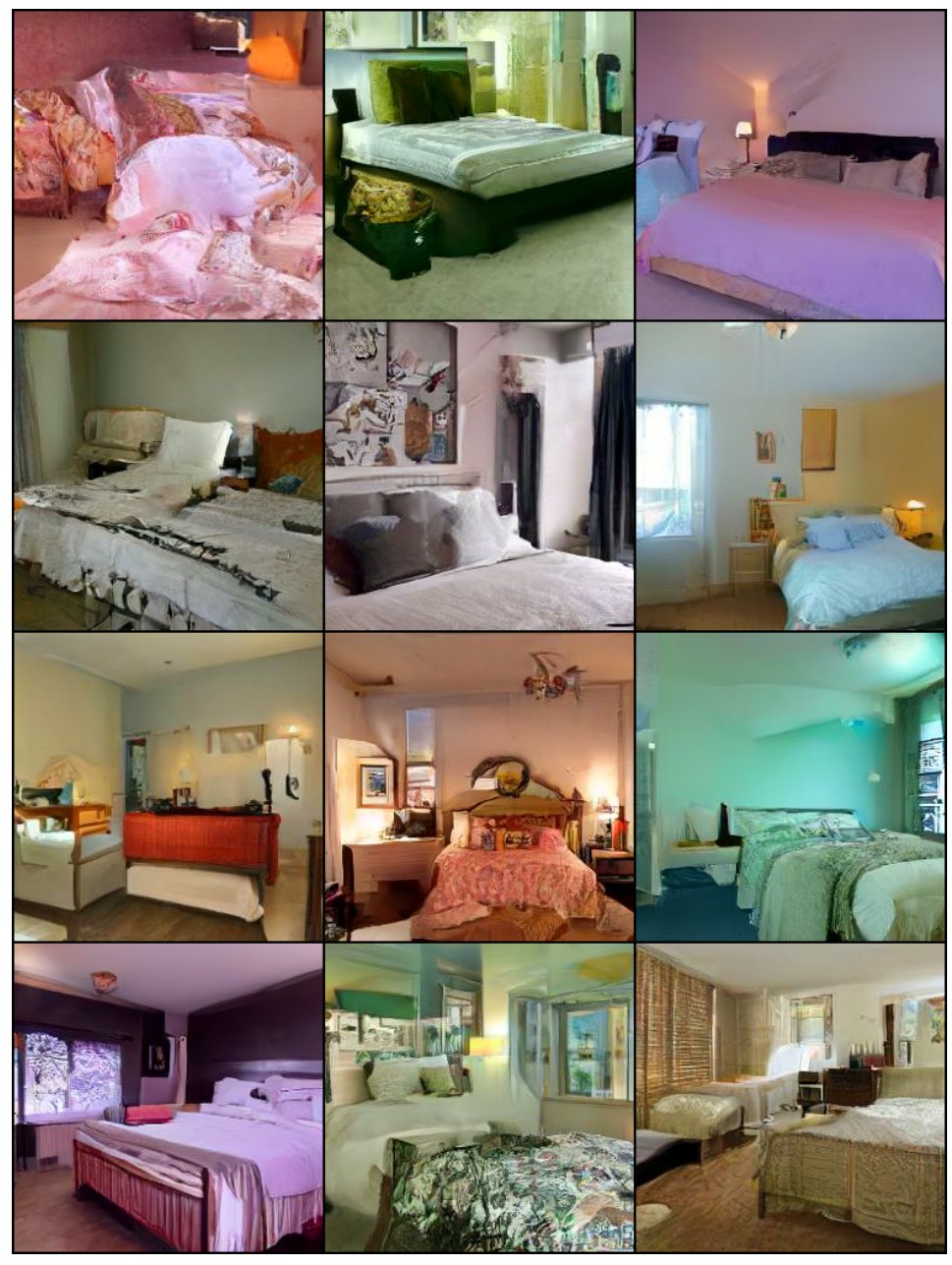

Figure 14: LSUN bedroom $256 \times 256$ samples from PFGM (RK45) using DDPM channel configuration.