# OpenReview forum: "Poisson Flow Generative Models"
_NeurIPS.cc/2022/Conference — NeurIPS 2022 Accept_

### Official Review · Reviewer_RM6S · 2022-07-06

**Rating:** 6
**Confidence:** 2
**Soundness:** 3 good
**Presentation:** 3 good
**Contribution:** 2 fair

**Summary:**

This paper propose a generative model based on Poisson equation. The authors augment each data point with a scalar variable z and model the true data point as points centered as $z = 0$  hyperplane. During training, the authors define a empirical Poisson field based on the current batch of data and run the forward ODE to calculate the normalized field $V_{B_L}$. They train the model $f_\theta$ to regress on $V_{B_L}$ via L2 loss. During the backward process, the author start from a large $z = z_{max}$ and graduately walk back to $z=0$ to generate samples. Using this model, the authors try to solve the problem in the original VE/VP ODES that the $l2$ norm of perturbed training data and the standard variance of Gaussain perturbation kernel may have strong correlation. They use various experiments to demonstrate the performance and effectiveness of the propose model.

**Questions:**

1. The sample quality: The authors claim that their model achieves comparable and even better performance as [1, 2] based on the scores reported in Table 1 and 2. However, the samples shown in Figure 3 seems to be worse than that shown in [1]. For example, comparing Figure 3 with Figure 14 in [1], it seems that the face samples in Figure3b contain more obvious artifacts. Do you have any explanation for that?
Also, the current paper only shows 64 x 64 samples while [1,2] can do 256 x 256.
Thus, it might be hard to judge whether the proposed method does improve the baseline.

2. The sampling time: Table1 and Table2 compare the sampling time through NFE. But since PFGM uses different ODE equation, would it be any time difference for runing each ODE step? Or would you please also comparing the total sampling time (maybe based on wall-clock time)?

3. The sensitivity of the hyperparameters: The proposed model introduce several hyperparametes like MAX_STEP, $z_{min}$, $z_{max}$, etc. How sensitive is the model if we change these parameters. Especially, Theorem 1 assumes $r \rightarrow \infty$. Then do we need to  carefully choose $z_{max}$ to meet this requirement?

[1] Improved techniques for training score-based generative models.

[2] Score-Based Generative Modeling through Stochastic Differential Equations


**Limitations:**

The authors have adequately addressed the limitations and potential negative societal impact of their work.

**Strengths And Weaknesses:**

For the strength of the paper:

I think this model proposed a new form of the SDE/ODE based generative model. By using Poisson's equation, the proposed PFGM model may have weaker $norm-z(t')$ correlation and propose good samples with less computation. The paper is well written and the ideas are easy to follow.

For the weakness:
1. The first weakness of the proposed method is the large computation needed during training. As shown and discussed in Limitations and Future Directions in the appendix, using more complex Poisson field, the PFGM is more than 2 times slower than the score-based models. This may limit the model from going to high-resolution and high-quality data.

2. Also as pointed out by the authors, besides the training time, the model defines an empirical Poisson field based only on a subset of the data, which means at each step, the  model $f_\theta$ is trying to fit a different underlying distribution and the estimation is biased. Although the authors try to mitigate the problem by using larger batch size, as the image space is highly multimodal, just enlarging batch size is not the final solution. This might also prevent the model from scaling up.
3. I also have some questions, please check the next part.

---

> ### Author Response · Authors · 2022-08-02
> **Thank you for your review and suggestions (Response part I)**
>
> Thank you for the detailed review and thoughtful feedback. Below we address specific questions.
>
> **Q: The first weakness of the proposed method is the large computation needed during training. As shown and discussed in Limitations and Future Directions in the appendix, using a more complex Poisson field, the PFGM is more than 2 times slower than the score-based models. This may limit the model from going to high-resolution and high-quality data.**
>
> A: We adopted a simpler perturbation algorithm as an alternative to forward ODE in the revised version of the paper (see new Algorithm 2). The perturbation algorithm adds noise to the augmented data to construct training samples. We observe that the new training scheme has the same training time as score-based models without hurting empirical performance. In addition, when paired with a more advanced architecture DDPM++, PFGM achieves record-breaking Inception/FID scores $9.62/2.54$ on CIFAR-10 (best in class in the continuous normalizing flow family). These are shown in Table 1 in the revised version.
>
> **Q: As pointed out by the authors, besides the training time, the model defines an empirical Poisson field based only on a subset of the data, which means at each step, the model $f_θ$ is trying to fit a different underlying distribution and the estimation is biased. Although the authors try to mitigate the problem by using larger batch size, as the image space is highly multimodal, just enlarging batch size is not the final solution. This might also prevent the model from scaling up.**
>
> A: Thanks for noticing the issue. We want to emphasize that the larger batch is only used for calculating the empirical Poisson field target during training. It is true that the expectation of this does not equal the full dataset empirical field. However, a large enough batch size provides a stable target for the model. We agree with the reviewer that there could be better solutions to further reduce the bias. However, PFGM already provides good sample quality when using batch size to 4096 (CIFAR-10 32 $\times$ 32), 256 (CelebA 64 $\times$ 64), 64 (LSUN bedroom 256 $\times$ 256) on DDPM++. The results suggest that bias is relatively small at these sizes.
>
> **Q: The sample quality: The authors claim that their model achieves comparable and even better performance as [1, 2] based on the scores reported in Table 1 and 2. However, the samples shown in Figure 3 seem to be worse than that shown in [1]. For example, comparing Figure 3 with Figure 14 in [1], it seems that the face samples in Figure 3b contain more obvious artifacts. Do you have any explanation for that?**
>
> A: Thanks for pointing out the issue. We hypothesize that the diversity of samples has larger effects on the FID scores than the artifacts. As shown in Fig 9(a) and Fig 9(b) in Appendix H.1.2, samples generated by PFGM have more diverse background colors and hair colors than samples of [1], when both methods use the NCSNv2 architecture.
>
> In addition, we evaluate the performance of PFGM on the DDPM++ architecture. We show that the FID score can be further reduced to $4.04$ using the more advanced DDPM++ architecture. By examining the generated samples of PFGM on DDPM++ (Fig 13 in Appendix I), we observe that the samples are diverse and exhibit fewer artifacts than PFGM w/ NCSNv2. It suggests that by using a more powerful architecture like DDPM++, we can remove the artifacts while retaining the diversity in PFGM.
>
> We have included the discussions above in Appendix H.1.2 in our new version.
>
> **Q: The current paper only shows 64 $\times$ 64 samples while [1,2] can do 256 $\times$ 256. Thus, it might be hard to judge whether the proposed method does improve the baseline.**
>
> A: As suggested by the reviewer, we experimented with PFGM (w/ DDPM++) on a higher resolution dataset LSUN bedroom 256 $\times$ 256. We have included the samples from PFGM in Appendix I. The generated samples in Fig. 14 have rich detail and clear shapes. It suggests that PFGM is able to scale up to high resolution datasets.

---

> > ### Author Response · Authors · 2022-08-02
> > **Response part II**
> >
> > **Q: The sampling time: Table1 and Table2 compare the sampling time through NFE. But since PFGM uses different ODE equations, would there be any time difference for running each ODE step? Would you please also compare the total sampling time (maybe based on wall-clock time)?**
> >
> > A: Thanks for the question. The main bottleneck of sampling time in each ODE step is the function evaluation of the neural network. Hence, for different ODE equations using similar neural network architectures, their inference times per ODE step are approximately the same.
> >
> > We implement PFGM on the NCSNv2 [1], DDPM++ [2], and DDPM++ deep [2] architectures, with some slight changes to account for the extra dimension $z$. In the table below, we report the sampling time per ODE step method with the DDPM++ backbone, as well as the total sampling time. We measure the sampling time of generating a batch of 1000 images on CIFAR-10. We compare PFGM, VP/sub-VP ODEs using the RK45 solver. As a reference, we also report the results of VP-SDE using the predictor-corrector sampler [2]. All the numbers are produced on a single NVIDIA A100 GPU.
> >
> >
> > |  Method      | NFE | Wall-clock time per step (second)     | Total wall-clock time (second)     |
> > |    :-    |    :----:   |          :---: |    :---: |
> > | PFGM (RK45)      | 110    | 0.526   | 57.81 |
> > | VP-ODE (RK45)   | 134      | 0.522      | 69.97 |
> > | sub-VP-ODE (RK45)     | 146       | 0.520   | 75.92 |
> > | VP-SDE (PC)     | 1000      | 0.491   | 490.65 |
> >
> > As expected, ODEs using similar architectures and the same solver have nearly the same wall-clock time per ODE step. The table also shows that PFGM achieves the smallest total wall-clock sampling time. We have included the results in Appendix H.2 in our revised version.
> >
> >
> > **Q: The sensitivity of the hyperparameters: The proposed model introduces several hyperparameters like MAX_STEP, $z_{min}$, $z_{max}$ etc. How sensitive is the model if we change these parameters. Especially, Theorem 1 assumes $r \to \infty$. Then do we need to carefully choose $z_{max}$ to meet this requirement?**
> >
> > A: Thanks for asking. As in [2], we set the terminal value of $z_{min}=1e-3$. We initially chose MAX_STEP ($M$ in the revised version) and $z_{max}$ that are large enough to satisfy the condition $\kappa \gg 1$ by the multipole expansion analysis in Appendix B. The condition guarantees that the data distribution can be roughly viewed as a point source at origin. For example, if we set $z_{max}=40$ on CIFAR-10, the corresponding $\kappa$ is greater than $50$ with high probability. The hyperparameters work well without further fine tuning. Hence, we hypothesize that PFGM is insensitive to the choice of hyperparameters in a reasonable range.
> >
> > We validate our hypothesis by varying $z_{max}$ during sampling. We report the FID scores by PFGM w/ DDPM++ architecture on CIFAR-10.
> >
> > |  $z_{max}$      | FID score |
> > |    :-    |    :----:   |
> > | 30     | 2.54    |
> > |40 |2.54 |
> > |50 |2.53  |
> >
> > We observe that the FID scores remain consistent with varying $z_{max}$. It is worth noting that the condition $\kappa \gg 1$ holds true for $z_{max}$ values ranging from 30 to 50. We have added the results and discussion to Appendix E.2.1 in our revised version.
> >
> >
> > *[1] Yang Song and Stefano Ermon. Improved techniques for training score-based generative models. ArXiv, abs/2006.09011, 2020.*
> >
> > *[2] Yang Song, Jascha Sohl-Dickstein, Diederik P. Kingma, Abhishek Kumar, Stefano Ermon, and Ben Poole. Score-based generative modeling through stochastic differential equations. ArXiv,abs/2011.13456, 2021*

---

> > ### Comment · Reviewer_RM6S · 2022-08-05
> > **Any quantitative results for the 256 x 256 experiments?**
> >
> > Thank you for your reply! And thank you for the efforts in the new experiments.
> > Would you please also provide the quantitative results (FID) for the newly added 256 x 256 experiments? And it would be the best if you can also compare it with some baseline scores. Because now by comparing the samples show here and the one shown in the DDPM[1] paper. It stills seems that the samples comes from PFGM contains more artifacts. Besides, I would also like to learn about the total training time for the 256 x 256 model. I know it could be hard to achieve SOTA results in such a short rebuttal period. But with the current results, I'm still a little bit concerned about the scale-up of PFGM to high resolution images (in the sense of both training cost and sample quality).
> >
> > [1] Jonathan Ho, etc. Denoising Diffusion Probabilistic Models

---

> > > ### Author Response · Authors · 2022-08-09
> > > **Thanks for the questions**
> > >
> > > Thanks for asking. We adopt the code base of [1] (https://github.com/yang-song/score_sde_pytorch) in our experiments. In paper [1], they experimented on the LSUN bedroom 256$\times$ 256 dataset only on VE-SDE using a deeper NCSN++ backbone. In our DDPM++ architecture, We directly borrow the configuration of channels from the NCSN++ architecture [1] in each residual block (PFGM w/ NCSN++ channel). The training time is approximately the same as NCSN++. It takes about 3.3 hours to finish 10k training iterations on two NVIDIA A100 GPUs. Due to the computational and time constraints, we only finished 500k of the total 2.4M training iterations before the Author Rebuttal deadline, and 800k training iterations before the Author-Reviewer Discussion deadline. We further change $z_{max}$ to $100$, as it empirically gives better sample quality.
> > >
> > > The FID scores steadily improve as training proceeds. We report the FID score of the latest PFGM checkpoint (800k iteration) using the NCSN++ channel configuration. We also evaluate the performance when using the configuration of channels in the DDPM [2] architecture (PFGM w/ DDPM channel), and complete 150k of the total 2.4M training iterations before the discussion deadline. Since it takes about 3 hours to generate 200 images by VE-SDE on two A100 GPUs, we only obtain 4k generated images of VE-SDE before the Author-Reviewer Discussion deadline. For fair comparison, we report the FID score calculated on 4k samples below using the evaluation protocol in https://github.com/openai/guided-diffusion/tree/main/evaluations .
> > >
> > > |  Method      | FID score (4k samples) | NFE    |
> > > |    :-    |    :----:   |          :---: |
> > > | VE-SDE      | 17.14   | 2000 |
> > > | PFGM w/ NCSN++ channel (800k iteration)    | 23.01    | 128 |
> > > | PFGM w/ DDPM channel (150k iteration)   | 19.47 | 134 |
> > >
> > > The table shows that PFGM still has comparable performance with VE-SDE, while achieving around 15$\times$ acceleration. We observe that PFGM achieves a better FID score using the similar configuration in the DDPM model, even though it only finishes 150k over the total 2.4M training iterations. Remarkably, the VE-ODE baseline — the method most comparable to ours — only produces noisy samples on this dataset. Empirically, we find that the FID score improves with more training iterations, and we can expect lower FID scores for the PFGM models as training proceeds. It suggests that PFGM is able to scale up to high resolution images when using advanced architectures.
> > >
> > > We also compare with DDPM [2] using similar architecture. The FID score calculated on 50k samples is **6.36** for the DDPM, as reported in [2], and **12.72** for PFGM w/ DDPM channel at the 150k training iteration checkpoint. Note that the DDPM requires **1000 NFE** during sampling, and doesn’t possess invertibility compared to flow models.
> > >
> > > We have included the samples generated by PFGM w/ DDPM channel at the 150k training iteration checkpoint in Fig.15 in our revised paper. The samples generated by PFGM w/ DDPM channel at checkpoint 150k and $z_{max}=100$ (Fig.15) appear cleaner and have fewer artifacts compared to samples generated by the original PFGM w/ NCSN++ channel at checkpoint 500k and $z_{max}=50$ (Fig.14). We will include the final quantitative results in our paper after completing the training of PFGM.
> > >
> > > *[1] Yang Song, Jascha Sohl-Dickstein, Diederik P. Kingma, Abhishek Kumar, Stefano Ermon, and Ben Poole. Score-based generative modeling through stochastic differential equations. ArXiv,abs/2011.13456, 2021*
> > >
> > > *[2] Jonathan Ho and Ajay Jain and P. Abbeel, Denoising Diffusion Probabilistic Models. ArXiv, abs/2006.11239*

---

> > > > ### Comment · Reviewer_RM6S · 2022-08-10
> > > > **Thank you for your response**
> > > >
> > > > Dear authors:
> > > >
> > > > Thank you for your response.
> > > > I decide to increase my rating to weak accept after reading your new response.
> > > >
> > > > Best,
> > > > Reviewer RM6S

---

### Official Review · Reviewer_cmyU · 2022-07-12

**Rating:** 4
**Confidence:** 4
**Soundness:** 2 fair
**Presentation:** 2 fair
**Contribution:** 2 fair

**Summary:**

The manuscript proposes using the Poisson equation in higher dimension to use as a generative model. Several numerical case studies are reported to illustrate the benefit of the proposed model in terms of computational speed and accuracy.

**Questions:**

- What is the mathematically rigorous justification of making the time derivative of $\boldsymbol{x}$  proportional to $\boldsymbol{E}(\boldsymbol{x})$? Can you do away with the heuristic argument in lines (84)-(85) and give mathematical arguments? Otherwise, one can think of infinitely many generative models (why just hemisphere?) and infinitely many vector fields for the same geometry of the Poisson equation.



**Limitations:**

The authors have stated some limitations. However, the main limitation I see, that is, the mathematically rigorous motivation/justification of the proposed method, is not discussed. I would have expected the authors to acknowledge this and put it in the Section I.

**Strengths And Weaknesses:**

- The paper proposes an interesting physics-motivated heuristic for generative models. Despite the initial idea being interesting, the reviewer gets the feeling that the authors did not spend efforts to make the ideas rigorous instead focused more on showcasing the idea on some examples through more fixes and heuristics along the way.

- Typos and carelessness galore. Some examples: $G(\boldsymbol{x},\boldsymbol{y})$ in equation (2) is valid only if the dimension $N\geq 3$, otherwise (for $N=2$) the formula is not valid and we need a logarithmic fundamental solution. The manuscript did not state this assumption on $N$. Line 191: "radical projection" --> "radial projection". In line 86: the positive function $f(\boldsymbol{x})$ is introduced without any other restrictions. We cannot guarantee Lipschitzness of the vector field $f(\boldsymbol{x})\boldsymbol{E}(\boldsymbol{x})$, and hence the existence-uniqueness of the forward and backward flows without any assumptions whatsoever. Notice that here regularity of $\boldsymbol{E}$ inherited from $\varphi$ which is in turn inherited from the regualrity of $\rho$, is not enough for the forward and backward ODE solutions' existence-uniqueness. In line 108: $p(\boldsymbol{x})$ is introduced as a probability distribution while in line 109 it is equated with $\rho(\boldsymbol{x})$ which was defined to be a density in line 68.

- The proofs, like the writing throughput the paper, are all heuristic lacking mathematical rigor and ignores all technical issues similar to the above. For instance, the crucial issue of whether the Poisson equation is guaranteed to have unique $C^2$ solution is not treated.

---

> ### Author Response · Authors · 2022-08-02
> **Thank you for your review and suggestions (Response Part I)**
>
> Thank you for the detailed review and thoughtful feedback. Before we address specific questions, we would like to briefly provide some context for our contribution within the recent literature and challenges in generative models.
>
> The series of recent works most similar to ours is the family of diffusion [1] and scored-based models [2]. Compared to other generative models such as GANs, diffusion models are more stable to train (not formulated as games) and have recently achieved impressive performance on various generative tasks such as image generation [1,2,3], text generation [4], molecular conformation generation [5,6], 3D point cloud generation [7]. However, diffusion models are somewhat slow to sample from. For instance, it took $1000 \sim 2000$ steps (image transformations) to generate high quality samples by the reverse SDE samplers on the CIFAR-10 image dataset [2]. [2] proposes backward ODE samplers (normalizing flow) that speed up the sampling process but these methods have not yet performed on par with the SDE counterparts.
>
> **Compared to diffusion models and score-based models, our PFGM offers two advantages.** First, PFGM's backward ODE sampler performs competitively with SDE methods while providing $10\times$ to $20\times$speed up across datasets. Second, our backward ODE delivers better generation performance than previous backward ODEs such as those of VE/VP/sub-VP SDEs [2]. In particular, our new simpler version achieves state-of-the-art sample quality in the normalizing flow family as measured by FID and Inception scores. Specifically, PFGM achieves a Inception score 9.68 and a FID score 2.48 on CIFAR-10.
>
> *[1] Jonathan Ho and Ajay Jain and P. Abbeel, Denoising Diffusion Probabilistic Models. ArXiv, abs/2006.11239*
>
> *[2] Nichol, A., Dhariwal, P., Ramesh, A., Shyam, P., Mishkin, P., McGrew, B., Sutskever, I., and Chen, M. GLIDE: Towards Photorealistic Image Generation and Editing with Text-Guided Diffusion Models. ArXiv, abs/2112.10741.*
>
> *[3] Yang Song, Jascha Sohl-Dickstein, Diederik P. Kingma, Abhishek Kumar, Stefano Ermon, and Ben Poole. Score-based generative modeling through stochastic differential equations. ArXiv,abs/2011.13456, 2021*
>
> *[4] Xiang Lisa Li and John Thickstun and Ishaan Gulrajani and Percy Liang and Tatsunori Hashimoto, Diffusion-LM Improves Controllable Text Generation. ArXiv, abs/2205.14217*
>
> *[5] Minkai Xu and Lantao Yu and Yang Song and Chence Shi and Stefano Ermon and Jian Tang, GeoDiff: a Geometric Diffusion Model for Molecular Conformation Generation. ArXiv, abs/2203.02923*
>
> *[6] Learning Gradient Fields for Molecular Conformation GenerationChence Shi and Shitong Luo and Minkai Xu and Jian Tang, ICML, 2021*
>
> *[7] Shitong Luo and Wei Hu, Diffusion Probabilistic Models for 3D Point Cloud Generation, 2021 IEEE/CVF Conference on Computer Vision and Pattern Recognition (CVPR), pages 2836-2844*
>
> **Q: G(x,y) in equation (2) is valid only if the dimension $N\ge 3$, otherwise (for $N=2$) the formula is not valid and we need a logarithmic fundamental solution. The manuscript did not state this assumption on $N$.**
>
> A: Thanks for pointing this out. We added the $N=2$ case in a footnote at the bottom of page 2. Note, however, that $N$ is typically much larger in the relevant applications. For example, it is equal to the number of pixels in an image, e.g., $N=3072$ in CIFAR-10.

---

> > ### Author Response · Authors · 2022-08-02
> > **Response Part II**
> >
> > **Q: In line 86: the positive function $f(\mathbf{x})$ is introduced without any other restrictions. We cannot guarantee Lipschitzness of the vector field $f(\mathbf{x})\mathbf{E}(\mathbf{x})$, and hence the existence-uniqueness of the forward and backward flows without any assumptions whatsoever. Notice that here regularity of $\mathbf{E}$ inherited from $\varphi$ which is in turn inherited from the regularity of $\rho$, is not enough for the forward and backward ODE solutions' existence-uniqueness.**
> >
> > A:  We motivated the Poisson flow more intuitively but agree with the reviewer that the relevant assumptions should be stated. The existence-uniqueness of the Poisson equation can be guaranteed by assuming that $\rho \in C^0$ and the support $\textrm{supp}(\rho)$ is a compact set. In Lemma 1, we show that the Poisson equation has a unique solution $\varphi \in C^2$ with zero boundary condition at infinity. We have explicitly stated the conditions in Section 2. We have also shown the existence-uniqueness of the Poisson equation in Lemma 1 and Lemma 2 in Appendix A in our revised version.
> >
> > Next, adding the condition $f \in C^1$ can guarantee the existence-uniqueness of the ODE $\frac{\partial \mathbf{x}}{\partial t}=f(\mathbf{x})\mathbf{E}(\mathbf{x})$. Since the solution to the Poisson equation satisfies $\varphi(\mathbf{x}) \in \mathcal{C}^2$, $\mathbf{E}(\mathbf{x}) =- \nabla_\mathbf{x} \varphi(\mathbf{x})$ belongs to $\mathcal{C}^1$. By the uniqueness and existence theorem of first-order ODE (Theorem 2.8.1 in [1]), we should have $f(\mathbf{x})\mathbf{E}(\mathbf{x}) \in \mathcal{C}^1$ so that ODE $\frac{\partial \mathbf{x}}{\partial t}=f(\mathbf{x})\mathbf{E}(\mathbf{x})$ has a unique solution. Thus $f(\mathbf{x})$ should be at least $\mathcal{C}^1$ since $\mathbf{E}(\mathbf{x}) \in \mathcal{C}^1$. We have added conditions that $f \in \mathcal{C}^1, \rho \in \mathcal{C}^0$ and $\textrm{supp}(\rho)$ is a compact set in Section 2.
> >
> > *[1] Henry J. Ricardo, in A Modern Introduction to Differential Equations (Third Edition), 2021*
> >
> > **Q: In line 108:  $p(\mathbf{x})$ is introduced as a probability distribution while in line 109 it is equated with $\rho(\mathbf{x})$ which was defined to be a density in line 68.**
> >
> > A: Thanks for asking. We want to clarify that a probability distribution $p(\mathbf{x})$ is a special case of "charge density" $\rho(\mathbf{x})$. Unlike charge density, $p(\mathbf{x})$ needs to be non-negative and integrate to unity. The background in Section 2 is applicable to densities (including probability distributions). From Section 3 onward, we focus on applications to probability distributions of data, which is the objective to be modeled in generative modeling. We have tried to clarify this with a footnote at the bottom of page 3.
> >
> > **Q: The proofs are all heuristic lacking mathematical rigor and ignore all technical issues similar to the above. For instance, the crucial issue of whether the Poisson equation is guaranteed to have a unique $\mathcal{C}^2$ solution is not treated.**
> >
> > A: Thanks for highlighting this. We now show in Lemma 1 (Appendix A) that the Poisson equation has a unique solution in $\mathcal{C}^2$ (up to a constant function) with zero boundary condition at infinity, given that $\rho \in \mathcal{C}^0$ and $\textrm{supp}(\rho)$ is a compact set. Our proof closely follows uniqueness theorems in electrostatics: we assume there are two different solutions, and derive that they can only differ by a constant. On the other hand, it is easy to show that the solution exists because the extension of Green’s function in $N$ dimensional space ($N\ge 3$) provides an explicit construction (Eq.(2)).
> >
> > Further, we prove the existence and uniqueness of the Poisson equation when $\rho$ is the data distribution in the augmented space in Lemma 2 (Appendix A), using the setup in Section 3. We have included the conditions of Lemma 2 in our proof of Theorem 2 (Appendix A), and added more technical details.

---

> > > ### Author Response · Authors · 2022-08-02
> > > **Response part III**
> > >
> > > **Q: What is the mathematically rigorous justification of making the time derivative of $\mathbf{x}$ proportional to $\mathbf{E}(\mathbf{x})$? Can you do away with the heuristic argument in lines (84)-(85) [the “honey” analogy] and give mathematical arguments?**
> > >
> > > A: We have rewritten the main text so that it avoids physics jargon and deferred physics discussions to Appendix J.
> > >
> > > From a mathematical perspective, the flow model is defined by the Poisson field, where the probability distribution evolves according to the gradient flow $\frac{\partial p_t(\mathbf{x})}{\partial t}=-\nabla\cdot(p_t(\mathbf{x})\mathbf{E}(\mathbf{x}))$. We can think of $p_t(\mathbf{x})$ as represented by a population of particles. The gradient flow is a special case of the Fokker-Planck equation, where the diffusion coefficient is zero [1]. The corresponding first-order ODE $d\mathbf{x}/dt = \mathbf{E}(\mathbf{x})$ is a special (non-diffusion) case of the Itô process. We can interpret the trajectories of the ODE as particles moving according to the Poisson field $\mathbf{E}(\mathbf{x})$, with initial states drawn from $p_0$.
> > >
> > > We have included the discussion in Section 2.
> > >
> > > From a physics perspective, the first-order dynamics can be realized in the overdamped limit. Newton's law implies that the external force is proportional to the acceleration of the particle. In the overdamped limit, e.g., when the particle is moving in honey, the external force is instead proportional to the velocity of the particle, making the equation of motion a first-order ODE. Denoting the viscosity of the fluid as $\gamma$, the dynamics of the particle under the influence of the electric field of the source $\rho(x)$ is
> > > $
> > >     m\frac{d^2\mathbf{x}}{dt^2}=-\gamma\frac{d\mathbf{x}}{dt} + q\mathbf{E}(\mathbf{x})
> > > $
> > > , which has an overdamped limit  $\frac{d\mathbf{x}}{dt}=q\mathbf{E}(\mathbf{x})$ when we set $t\to\gamma t$ and $\gamma\to\infty$. In this case, a particle with mass $m=1$ and charge $q=1$ would follow the electric field with a velocity equal to $\mathbf{E}$, justifying Eq. (4). We have added the physical discussion to Appendix J.
> > >
> > > *[1] Fokker–Planck equation. In Wikipedia. https://en.wikipedia.org/wiki/Fokker–Planck_equation*
> > >
> > > **Q: Why just hemisphere?**
> > > A: Poisson flow can map any data distribution to the uniform distribution on the infinite hemisphere (Theorem 1), which makes the hemisphere special.
> > >
> > >
> > > **Q: Typos**
> > >
> > > A: Thank you for the corrections for the typos. We have polished the writing according to your suggestions.

---

### Official Review · Reviewer_F5nf · 2022-07-14

**Rating:** 6
**Confidence:** 4
**Soundness:** 3 good
**Presentation:** 2 fair
**Contribution:** 3 good

**Summary:**

The authors of this paper propose a physics based generative model named Possion flow generative model (PFGM). In this model, the source function corresponds to the data distribution; and a dual ODEs serve as mapping between the data distribution and the positive hemisphere with radius r. The generative process starts from a point on the hemisphere, follows the negative electric field, and finally generate a sample. This model was tested on CIFAR-10 and CelebA and compared with state of the art image generation models including score-based methods and obtained comparable or better performance. Furthermore, it is shown that PFGM's generation process is much faster than existing methods.

**Questions:**

1. It is highly recommended that PFGM should be investigated on a third dataset.
2. It would be interesting to see a direct comparison with ViT based image generation models.

**Ethics Review Area:**

["I don’t know"]

**Limitations:**

Potential social impacts are discussed in the supplementary file.

**Strengths And Weaknesses:**

Strengths:
1. Inspired by Poisson equation in physics, the idea of this deep generative model is interesting. To avoid generated data points collapse at one point, augmented dimension is applied.
2. The generation process in PFGM is more efficient than existing models for image generation.

Weaknesses:
1. Quantitative analysis unveils that PFGM seems to work better on larger datasets. This should be furthered investigated on a third dataset.
2. The authors should discuss and compare (if possible) with ViT based generative model, e.g., https://arxiv.org/abs/2107.04589.

---

> ### Author Response · Authors · 2022-08-02
> **Thank you for your review and suggestions**
>
> Thank you for the detailed review and thoughtful feedback. Below we address specific questions.
>
> **Q: Quantitative analysis unveils that PFGM seems to work better on larger datasets. This should be further investigated on a third dataset.**
>
> A: PFGM can outperform previous state-of-the-art methods even on the CIFAR-10 dataset. We replaced NCSNv2 with more advanced DDPM++ [1] and adopted the simpler training algorithm (Algorithm 2). We report new results in Table 1 in our revised version. The results show that the backward ODE of PFGM achieves a record Inception score of 9.62 and a FID score of 2.54 on CIFAR-10 in the normalizing flow family. Thus PFGM can outperform previous methods using the same neural architectures even on smaller datasets.
> As suggested by the reviewer, we also conducted experiments on a higher resolution dataset, LSUN bedroom 256 $\times$ 256. We include the samples from PFGM in Appendix I. The generated samples have rich detail and clear shapes. We omit quantitative results for this dataset since these are not widely reported.
>
> **Q: The authors should discuss and compare (if possible) with ViT based generative model**
>
> A: Thanks for the suggestion. We have included a discussion of ViT based-generative models in our introduction. We also compare with the ViTGAN [2] model on CIFAR-10 in Table 1.
>
>
> *[1] Yang Song, Jascha Sohl-Dickstein, Diederik P. Kingma, Abhishek Kumar, Stefano Ermon, and Ben Poole. Score-based generative modeling through stochastic differential equations. ArXiv, abs/2011.13456, 2021*
>
> *[2] Kwonjoon Lee and Huiwen Chang and Lu Jiang and Han Zhang and Zhuowen Tu and Ce Liu, ViTGAN: Training GANs with Vision Transformers. ArXiv, abs/2107.04589*

---

> > ### Comment · Reviewer_F5nf · 2022-08-09
> > **Update**
> >
> > I would like to thank the authors for answering my and other reviewers' questions. I have read all these comments and responses. Within a limited period of rebuttal time, the authors were able to obtain some preliminary results on LSUN. ViTGAN was also included in the comparison. Overall, the idea of this paper is interesting. It can achieve comparable results with the current state of the art methods for image generation. For these reasons, I would like the increase my rating to 6 week accept.

---

### Author Response · Authors · 2022-08-02
**A summary of updates**

We would like to thank all the reviewers for their constructive feedback. We have revised our draft according to these comments. Major revisions are highlighted in blue in the new version. In particular, we show that PFGM achieves best in class performance in the normalizing flow family when combined with more advanced architectures. We have also added mathematical conditions in place of more intuitive ones in our revised version. Below we provide a brief summary of these updates:

## 1. Experiments on more advanced architectures
We adopted more advanced neural architectures DDPM++ [1] and DDPM++ deep [1]. We also replaced the slow forward ODE simulation with a simpler perturbation algorithm (Algorithm 2). We show experimentally that with these changes PFGM achieves state-of-the-art performance on the CIFAR-10 dataset among the normalizing flow models, with record FID/Inception scores of $\bf{2.54/9.62}$ when using DDPM++ and $\bf{2.48/9.68}$ with DDPM++ deep. In addition, the resulting method performs competitively with SDE samplers of state-of-the-art score-based models [1] while offering $10\times$ to $20\times$ speed up.

We have added the new results in Table 1 in the revised version.

## 2. More comparisons / experiments
In response to Reviewer F5nf and Reviewer RM6S, we have experimented on a higher resolution dataset, LSUN bedroom $256 \times 256$, to showcase the scalability of PFGM (Appendix I). As suggested by Reviewer F5nf, we have included the discussion (Section 1) and comparison (Table 1) of ViTGAN.

In response to Reviewer RM6S, we provided the wall-clock sampling time (Appendix H.2), the sensitivity analysis of hyperparameters (Appendix E.2.1), and the discussion of artifacts and diversity of the CelebA images (Appendix H.1.2).

## 3. Mathematically rigorous motivation/justification
As suggested by Reviewer cmyU, we have added relevant mathematical assumptions to keep the paper more self-contained. For example, we have included the technical conditions about the source term and the function $f$ needed to ensure the existence-uniqueness of the solution to the Poisson equation and the ODE (Section 2, Lemma 1 in Appendix A). We also added more rigor to the proof of theorem 1 (Lemma 2 and Theorem 2 in Appendix A). In addition, we replaced what we thought to be more intuitive arguments in Section 2 with mathematical arguments (Section 2, Appendix J).

## 4. Faster training algorithm
We replaced the slow forward ODE simulation with a simplified perturbation algorithm. The perturbation algorithm adds noise to the augmented data to construct training samples similar to score based models. The new training procedure has the same training time as score-based models (2 times faster than the old algorithm) without any degradation in model performance.

*[1] Yang Song, Jascha Sohl-Dickstein, Diederik P. Kingma, Abhishek Kumar, Stefano Ermon, and Ben Poole. Score-based generative modeling through stochastic differential equations. ArXiv,abs/2011.13456, 2021*

---

### Meta-Review · Area_Chair_vwEc · 2022-08-26

**Recommendation:** Accept
**Confidence:** Certain

**Metareview:**

 All the reviewers agreed that the paper is novel and interesting with significant contributions. While there were certain concerns regarding the experimentation, clarity, and mathematical rigor initially, the extensive rebuttal provided by the authors addressed most of the concerns, hence some reviewers increased their scores. Hence, I am happy to recommend an acceptance for the paper.

However, I must say that, I find the phrase "record breaking" academically inappropriate and I kindly request the authors to replace it with a more academic phrase, such as achieving the state-of-the-art.

**Award:**

No

---

### Decision · Program_Chairs · 2022-09-14

Accept